# 4D nanoimaging of early age cement hydration

Shiva Shirani[1], Ana Cuesta[1], Alejandro Morales-Cantero[1], Isabel Santacruz[1], Ana Diaz[2], Pavel Trtik[3], Mirko Holler[2], Alexander Rack[4], Bratislav Lukic[4], Emmanuel Brun[5], Inés R. Salcedo[6] & Miguel A. G. Aranda[1]✉

Despite a century of research, our understanding of cement dissolution and precipitation processes at early ages is very limited. This is due to the lack of methods that can image these processes with enough spatial resolution, contrast and field of view. Here, we adapt near-field ptychographic nanotomography to in situ visualise the hydration of commercial Portland cement in a record-thick capillary. At 19 h, porous C-S-H gel shell, thickness of 500 nm, covers every alite grain enclosing a water gap. The spatial dissolution rate of small alite grains in the acceleration period, ~100 nm/h, is approximately four times faster than that of large alite grains in the deceleration stage, ~25 nm/h. Etch-pit development has also been mapped out. This work is complemented by laboratory and synchrotron microtomographies, allowing to measure the particle size distributions with time. 4D nanoimaging will allow mechanistically study dissolution-precipitation processes including the roles of accelerators and superplasticizers.

Portland concrete is the world's largest fabricated commodity, ~20 billion tonnes/yr. The enormous production of Portland cement (PC), ~4 billion tonnes/yr, results in ~2.7 billion tonnes/yr of $CO_2$ emissions[1]. Therefore, there are many attempts to decrease the cement $CO_2$ footprint[1,2]. In order to rationally design approaches to decrease the embodied carbon content of binders, maintaining the performances, cement hydration understanding is key. Unfortunately, there are many unanswered questions[3] regarding the complex dissolution and precipitation processes that lead the setting and early hardening of cements[4].

The hydration of PC can be divided into five periods[3], see Fig. 1b. Stage-I is the initial dissolution (first minutes); stage-II is the low activity, induction, period (some hours); stage-III is the acceleration (several hours until the maximum of the heat flow trace); stage-IV is the deceleration (tens of hours); and stage-V is the diffusion-controlled hydration (months to years). There was a strong debate about the mechanism responsible of the induction period. However, it is now accepted that it is the dissolution controlled by

undersaturation[5] and not the protective membrane theory. Conversely, there is no agreement in the mechanism(s) to explain the transition from acceleration to deceleration, when there is a degree of hydration of just 10-20% and plenty of space for the hydrates to grow. The most advanced theories, recently discussed[6], are based on heterogeneous nucleation and growth within confined regions taking into account the initial particle size distributions[7], for instance, see the reaction zone hypothesis[8]. There are alternatives[9] like the needle model[6] where alite, the most abundant component of PC, hydrates[10,11] to yield nonstoichiometric calcium-silicate-hydrate (C-S-H) gel[12], which nucleates and grows as needles. Neither the dissolution of small grains, nor water diffusion, nor etch-pits coalescence, nor C-S-H gel impingement −alone− can currently explain the transition from the acceleration to the deceleration periods[3]. The factors affecting the C-S-H gel growth in these two periods, III and IV, are not known. Moreover, the role of etch-pits[13] needs to be better understood as well as the consequences of the spatial gap which opens between the dissolving (inward) alite grains and the growing (mainly

[1]Departamento de Química Inorgánica, Cristalografía y Mineralogía, Universidad de Málaga, 29071 Málaga, Spain. [2]Laboratory for Macromolecules and Bioimaging, Paul Scherrer Institut, 5232 Villigen PSI, Switzerland. [3]Laboratory for Neutron Scattering and Imaging, Paul Scherrer Institut, 5232 Villigen PSI, Switzerland. [4]ESRF-The European Synchrotron, 71 Rue des Martyrs, 38000 Grenoble, France. [5]Université Grenoble Alpes, Inserm UA7 STROBE, 38000 Grenoble, France. [6]Servicios Centrales de Apoyo a la Investigación, Universidad de Málaga, 29071 Málaga, Spain. ✉e-mail: g_aranda@uma.es

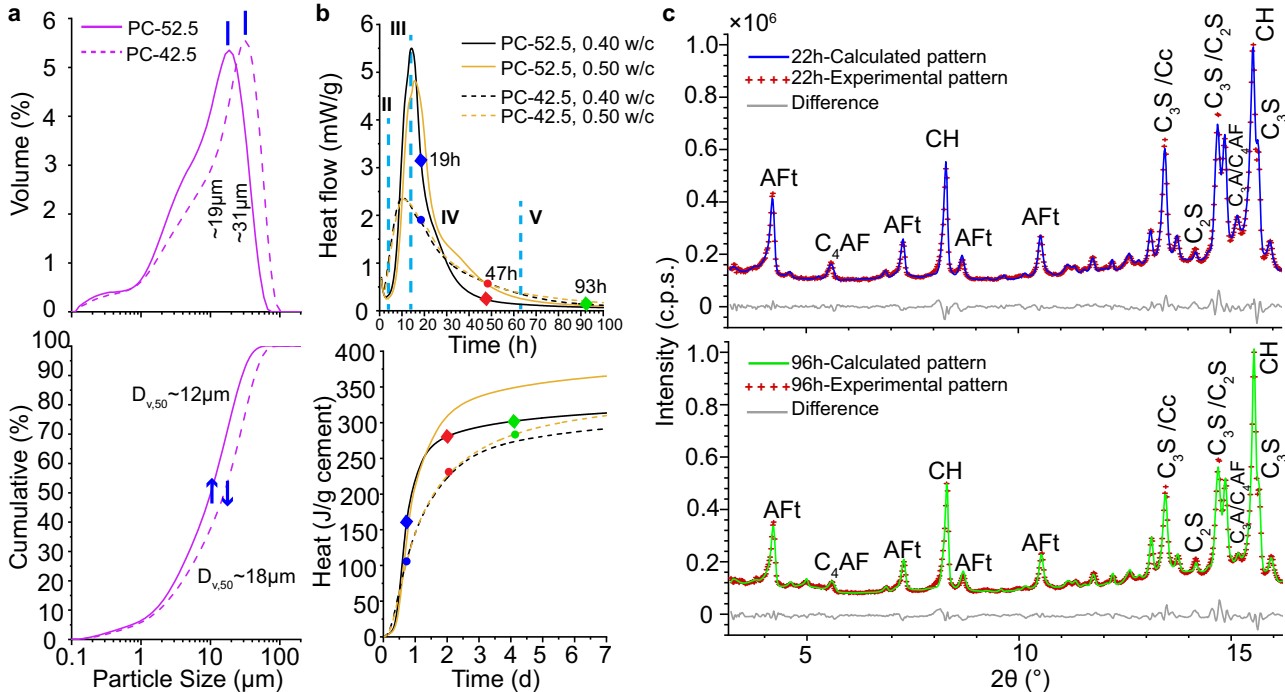

**Fig. 1 | Initial characterization of the employed Portland cements. a** Textural analysis of the two cements: PC-52.5 and PC-42.5, with Blaine values of 409 and 368 $m^2kg^{-1}$, respectively. (top) Particle size distributions, (bottom) cumulative volume variation as measured by laser scattering. **b** Isothermal calorimetric study, T = 25 °C, for the cement pastes prepared with w/c values of 0.50 and 0.40, referenced to 1 g of anhydrous cement. (Top) Heat flow curves (where the typical hydration stages are sketched), (bottom) cumulative heat traces for the two cements. **c** Laboratory Rietveld plots ($MoK\alpha_1$ radiation, $\lambda = 0.7093$ Å) for PC-52.5 paste, w/c = 0.40, within a capillary of 1 mm of diameter. The main diffraction peaks are labelled with the contributing crystalline component by using the cement notation: tetracalcium aluminoferrite ($C_4AF$), belite ($C_2S$), alite ($C_3S$), tricalcium aluminate ($C_3A$), calcite (Cc), portlandite (CH) and ettringite (AFt). (Top) Pattern collected at 22 h, (bottom) pattern collected at 96 h of hydration.

outward) C-S-H gel[14,15]. Finally, the density evolution of the C-S-H gel shells is also unknown.

On the one hand, in situ laboratory[16] and synchrotron[17] powder diffraction allow following the phase development with time. These studies yield volume-averaged information which misses any spatial feature like particle size dependence. On the other hand, electron microscopy (EM) techniques yield very valuable information, with high spatial resolution, but they only give snapshots, as the experimental conditions are not compatible with the hydration in relevant conditions. In situ tomography[18,19] can contribute to filling this gap. In cements, modelling[20] and microstructural characterization methods[21] acknowledge the growing importance of X-ray microtomography ($\mu$CT) in their different modalities[22]. Moreover, $\mu$CT is being widely used[23] and in particular to follow in situ 4D (3D + time) some specific features of cement hydration[24–33]. In the last three years, important advances have been reported including: i) the automated correction for the movement of suspended particles at very early ages[34] which allowed to follow in situ PC hydration after water mixing[35]; ii) to follow the fast dissolution of plaster and the precipitation of gypsum[36]; iii) the simultaneous use of neutron and laboratory X-ray tomographies for in situ studying the microstructural changes of PC mortars on moderate heating[37]; and iv) the measurement of alite particle dissolution using fast synchrotron nano X-ray computed tomography[38,39]. However, none of these 4D imaging works combine the stringent four requirements needed for carrying out relevant contributions to the understanding of the mechanism(s) of Portland cement hydration at early ages: (i) water to cement mass ratio (w/c) close to 0.50, (ii) submicrometer spatial resolution, (iii) good contrast to be able to identify the different evolving components (more than eight), and (iv) relatively large scanned volume to allow hydration to progress with appropriate particle sampling, the particle sizes of commercial PCs have $D_{v,50} \in 10-20$ $\mu$m. In particular, hard X-ray synchrotron microtomography has not the required submicrometer spatial resolution neither sufficient component contrast[35,40], hard X-ray synchrotron nanotomography has not the required contrast between the components to be able to classify the hydrates[38,39] and soft X-ray synchrotron nanotomography has the contrast but it requires very large w/c ratios and very small fields of view which does not allow the hydrates to growth in relevant condition (i.e. confined space with low water-cement ratios)[32].

So far, ptychographic X-ray computed tomography (PXCT)[41], which merges scanning X-ray microscopy and coherent diffraction imaging[42–44], met the first three requirements. Hence, it was applied to several binders within capillaries of ~40 $\mu$m of diameter[45,46] using a photon energy of 6.2 keV. The second study[46] provided valuable information about C-S-H gel hydrated for 5 months: an average stoichiometry of $(CaO)_{1.80}(SiO_2)(H_2O)_{3.96}$ with a mass density of 2.11 $gcm^{-3}$ and an electron density of 0.64 $e^-Å^{-3}$. Moreover, it allowed quantifying a 6.4 vol% of a second amorphous component, iron-siliceous hydrogarnet, with 2.52 $gcm^{-3}$ and 0.76 $e^-Å^{-3}$. Here, we have used PXCT in near-field configuration[44,47] to acquire data in a record-thick capillary of ~160 $\mu$m, employing a higher photon energy, 8.93 keV. This configuration, and the iterative algorithms that allowed the reconstructions[44], now meet simultaneously the four stringent requirements with current data collections of 3-4 h allowing 4D measurements of spatially resolved data during the first four days of cement hydration. On the one hand, the relatively slow overall acquisition time is the main limitation of this work, but this is expected to improve, see last section. On the other hand, the excellent spatial resolution and contrast of ptychographic nanotomography gave quantitative values of relevant parameters in the dissolution-precipitation processes like alite spatial dissolution rates

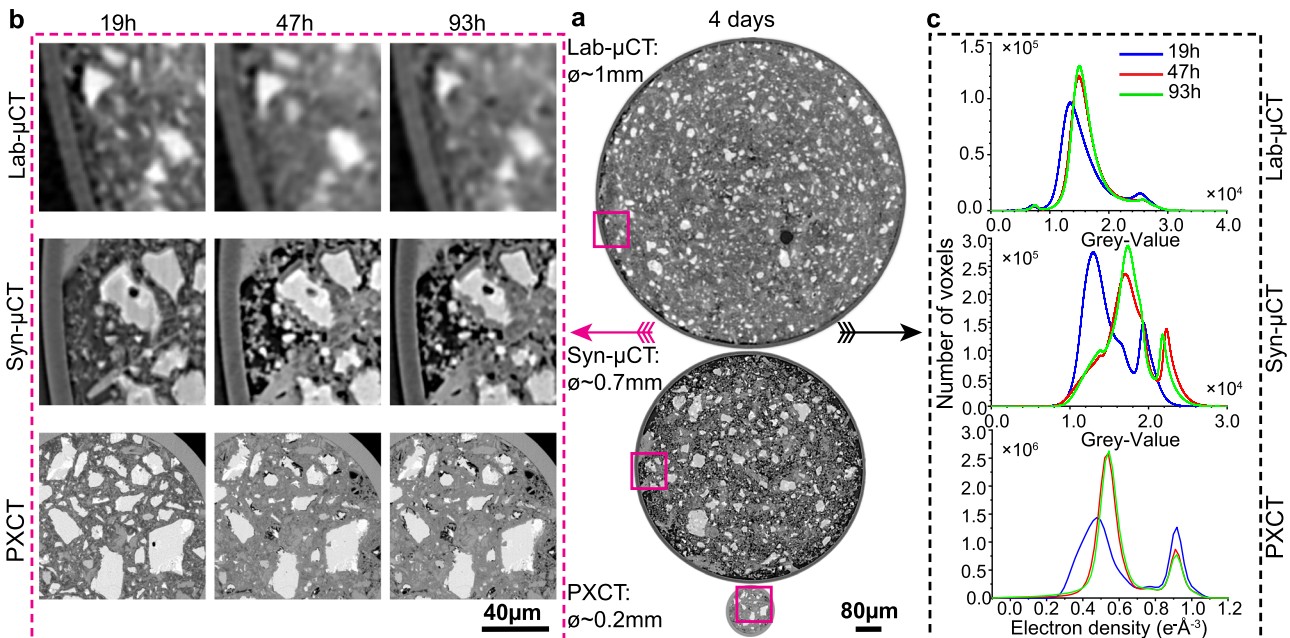

**Fig. 2 | In situ multicontrast X-ray tomographic studies of cement hydration.** **a** Selected orthoslices for the three imaging approaches. (Top) Attenuation-contrast laboratory data (Lab-μCT) for a PC-52.5 paste with w/c = 0.40. (Middle) Inline propagation-based phase-contrast synchrotron data (Syn-μCT) for a PC-42.5 paste with w/c = 0.50 [phases retrieved by using the Paganin algorithm[51]], (bottom) Quantitative phase-contrast, phases retrieved by near-field PXCT for a PC-52.5 paste scanning a region with w/c~0.40. The thicknesses of the capillaries are given. **b** Enlarged views for the three approaches, at the three hydration ages, to qualitatively illustrate the quality of the data (contrast and spatial resolution). Clinker minerals are seen as whitish particles, porosity as darkish regions, and hydrates have intermediate grey tones. **c** Histograms for the different tomographic studies and hydration times. The histograms were obtained by computing the largest possible volumes without including the glass capillary walls.

and etch-pit growth rate. These values can help to test the above described models.

## Result and discussion
### Initial cement analysis and cement hydration study

Two commercial PCs have been used and their laboratory X-ray powder diffraction (LXRPD) patterns were analysed by the Rietveld method, see Fig. S1. The elemental and mineralogical analyses are given in Tables S1 and S2, respectively. As reported in Table S2, the anhydrous cements are mixtures with more than eight crystalline phases. The cements have very similar elemental and mineralogical compositions but differ in their textural properties, see Fig. 1a and Table S3. The specific surface areas for PC-52.5 and PC-42.5 were 2.27 and 1.25 m²g⁻¹, respectively. PC-52.5, with finer particles $D_{v,50}$~12 μm, was used for the PXCT study in order to have more hydrating particles in the analysed volume. It is noted that, in a very recent nanotomographic study[7], researchers milled alite very extensively, all particles<10 μm, in order to fit them within a small field of view (FOV) of ~50 μm. PC-52.5 was used for the PXCT and laboratory μCT imaging studies and the additional laboratory characterization. PC-42.5, with slightly larger average particle size, was used for the synchrotron μCT imaging study.

The calorimetric study, see Fig. 1b, gave the hydration kinetic features at the relevant w/c ratios of 0.50 and 0.40. Table S4 reports the cumulative heat at the hydration times where the imaging data were acquired. Moreover, it also lists the degree of hydration (DoH) at those times to be used as a reference in the imaging studies; for a detailed explanation of this type of calculation, see ref. 48. As expected, PC-52.5 releases more heat than PC-42.5, mainly because of its finesse. The DoH at the maxima of the heat flow peaks were 19% and 10% for PC-52.5 and PC-42.5, respectively. These values are well-known[3,49,50] but they cannot be explained by current models.

Figure 1c displays the LXRPD Rietveld fits for hydrating PC-52.5 taken in the same capillary where the Lab-μCT imaging study was carried out. The in situ LXRPD data were analysed by the Rietveld method resulting in the quantitative phase analyses reported in Table S5. From these data, the DoHs of the different clinker phases are derived at 22, 50 and 96 h of hydration, also used as reference.

### In situ X-ray tomographic studies of cement hydration

Two additional in situ X-ray imaging studies were carried out to place the results of the PXCT nanoimaging in context. Emphasis is placed on the accuracy of the results that can only be estimated by comparison. The FOVs were cylinders. Figure 2a displays one orthoslice for each work: (i) Lab-μCT for PC-52.5-w/c = 0.40, FOV = 1200×940 μm (ϕ×L); (ii) Syn-μCT for PC-42.5-w/c = 0.50, FOV = 800 × 1190 μm; and (iii) PXCT for PC-52.5-w/c~0.40, FOV = 186 × 30 μm. Syn-μCT data has a larger w/c value, showing higher porosity (the darkish micro-regions) see the central panel in Fig. 2a. The nominal w/c mass ratio employed to fill the PXCT narrow capillary, 200 μm of nominal diameter, was 0.50, see methods. However, it is very difficult to accurately control the w/c ratio in very thin capillaries. Therefore, the w/c ratio of the scanned volume for the PXCT measurement was measured as previously published[46] and detailed in a subsection of the S.I. The w/c ratio of the scanned volume in the PXCT study was 0.41.

Figure 2b shows enlarged views to illustrate the evolution of the different components and the spatial resolutions. The PXCT study has a much higher resolution and contrast at the expense of a smaller FOV. A minor artefact can be seen in the Syn-μCT data as some borders have grey-values too white. This is a common feature for inline propagation-based data that cannot be fully corrected with the employed Paganin algorithm[51]. Figure 2c shows the histogram evolutions with time. At 19 h (blue traces), there is plenty of free water, which displaces the main peak toward smaller grey-values/electron densities. As hydration progresses, and as expected, the main peak in the histograms densifies and the amount of clinker components decreases. The histogram evolutions for the same cement paste from Lab-μCT and PXCT, see corresponding panels in Fig. 2c, were very similar giving confidence to

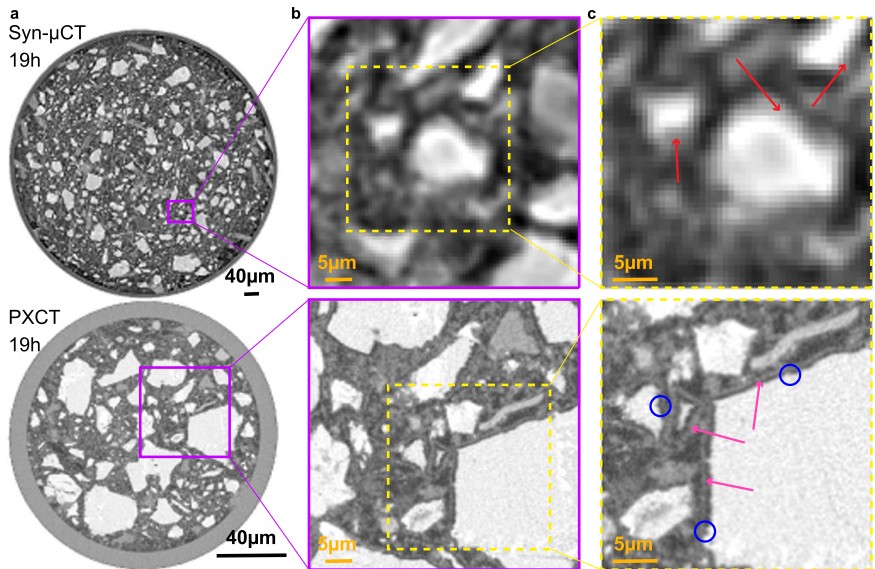

**Fig. 3 | Comparison of phase-contrast synchrotron tomography and ptychographic X-ray computed tomography. a** Selected orthoslices at 19 h for (top) Syn-μCT [voxel size: 650 nm, total scanned volume: 5.25 10⁸ μm³, overall acquisition time: 5 min], and (bottom) PXCT [voxel size: 186.64 nm, total scanned volume: 8.15 10⁵ μm³, overall acquisition time: 3 h, 55 min]. **b** Enlarged views of the highlighted regions (purple squares) in **a**, in order to illustrate the level of detail that can be observed with these imaging modalities. Every voxel in Syn-μCT image starts to be evident. **c** Further enlarged views to illustrate the maximum level of detail that can be observed. (Top) The Syn-μCT image shows whitish particles (anhydrous cement particles) surrounded by hydrates (greyish voxels) which are highlighted by red

arrows. (Bottom) The PXCT data, at the same magnification, shows a much higher level of detail. The C-S-H gel shells surrounding the alite particles are clearly visible (pink arrows). There is a water gap between the shell and the alite grain due to the inward dissolution of alite. Moreover, etch-pits on the surfaces of the alite particles are also visible (blue circles). The highest spatial resolution and better contrast of PXCT data allow visualizing submicrometre features of the dissolution-precipitation processes which are not visible in propagation-based Syn-μCT. Conversely, propagation-based Syn-μCT permits to scan much larger volumes in much smaller acquisition times, showing the complementary nature of both techniques.

the relevancy of the nanoimaging results in spite of the limited amount of volume scanned to have submicrometer resolution. The grey-values in the Lab-μCT study, see Fig. 2c (top panel), are related to the attenuation coefficients of the components in this PC-52.5 paste, but the relationship is not quantitative due to the polychromatic nature of the employed radiation. Conversely, the electron density values obtained for the same paste by PXCT are quantitative. Therefore, the grey scales in the Lab-CT and the electron densities in the PXCT datasets cannot be directly related as they derived from the imaginary and the real part of the refractive index of every component.

The spatial resolution was characterised by two approaches as recently reported[52]. The procedures are thoroughly detailed in Supplementary Information (SI). On the one hand, the spatial resolution can be determined by the edge sharpness across selected interfaces. A point spread function (PSF) used to determine the spatial resolution of the images as ISO/TS 24597 defines the Gaussian radius of the PSF as the resolution, which equals a change between 25%–75% grey value along the studied interfaces[38]. The spatial resolutions, determined by this approach, were 250(25) nm, 264(25) nm, 272(34) nm, 748(19) nm and 2.21(17) μm, for PXCT-19h, PXCT-47h, PXCT-93h, Syn-μCT and Lab-μCT datasets, respectively. As examples of this procedure, Figs. S2–S4 display line profiles of sharp interfaces between high (i.e. alite) and low density (i.e. porosity) components within the capillaries. On the other hand, Fourier-shell-correlation (FSC)[53] has also been employed to estimate spatial resolution. Figs. S5–S7 displays the FSC traces for the three imaging modalities. The agreement between both approaches is satisfactory for Syn-μCT and Lab-μCT, but not for PXCT. The worse spatial resolution estimated by FSC for PXCT is very likely due to the low number of projections, i.e. 420, which make the subtomograms employed in the FSC calculation severely undersampled. Moreover, Fig. 3 compares the level of details that can be obtained with Syn-μCT and PXCT at 19 h of hydration. The latter shows the C-S-H gel shell and a porosity gap between the shell and the dissolving alite particle that it

is not observed in the Syn-μCT data because the lack of spatial resolution and contrast.

## 4D nanoimaging of cement hydration

In situ near-field PXCT data were taken as detailed in methods. To ensure the relevance of the results, the scanned volumes were assessed. Firstly, the w/c ratio was 0.41(2), as determined from the absorption data[46], see Table S6. This w/c value is fully consistent with the obtained degrees of hydration. Secondly, possible signatures of radiation damage were explored. The mean electron density values of the whole sample were 0.600, 0.599 and 0.591 e·Å⁻³, for the 19, 47 and 93 h datasets, respectively. The spatial resolution from FSC was 470 and 500 nm, for the 47 and 93 h data, respectively. Hence, radiation damage cannot be discarded but it is small, if any. Thirdly, seven sets of components within the tomograms were identified: air, water, AFt/C-S-H gel/others, CH, Cc, $C_3A/C_3S/C_2S$ and $C_4AF$, using the electron density and the absorption data in the bivariate plots, see Figs. S8 and S9. The calculated electron density and attenuation length values are given in Table S7, and Fig. S10 displays the histogram evolution in logarithm scale. The differences between the theoretical electron densities and the measured ones are mainly due to partial volume effects. For instance, portlandite, i.e. $Ca(OH)_2$, has a theoretical electron density value of 0.69 e·Å⁻³. The measured values at 19 and 93 h were 0.62(2) and 0.651(5) e·Å⁻³, see Table S7. These numbers are 6-10% smaller than the theoretical one, with the difference being higher than the errors of the measurements, which are estimated in 2-3%[46,54]. This difference is very likely due to the presence of residual water porosity below the spatial resolution of the measurements, which we refer to partial volume effects. It should be noted that individual C-S-H nanoparticles have sizes close to 5 nm[12,55] much smaller than the spatial resolution of this work, i.e. ~250 nm. Therefore, the C-S-H regions analysed here very likely contain other components like gel and capillary water porosities and interspersed calcium hydroxide. It should also be noted

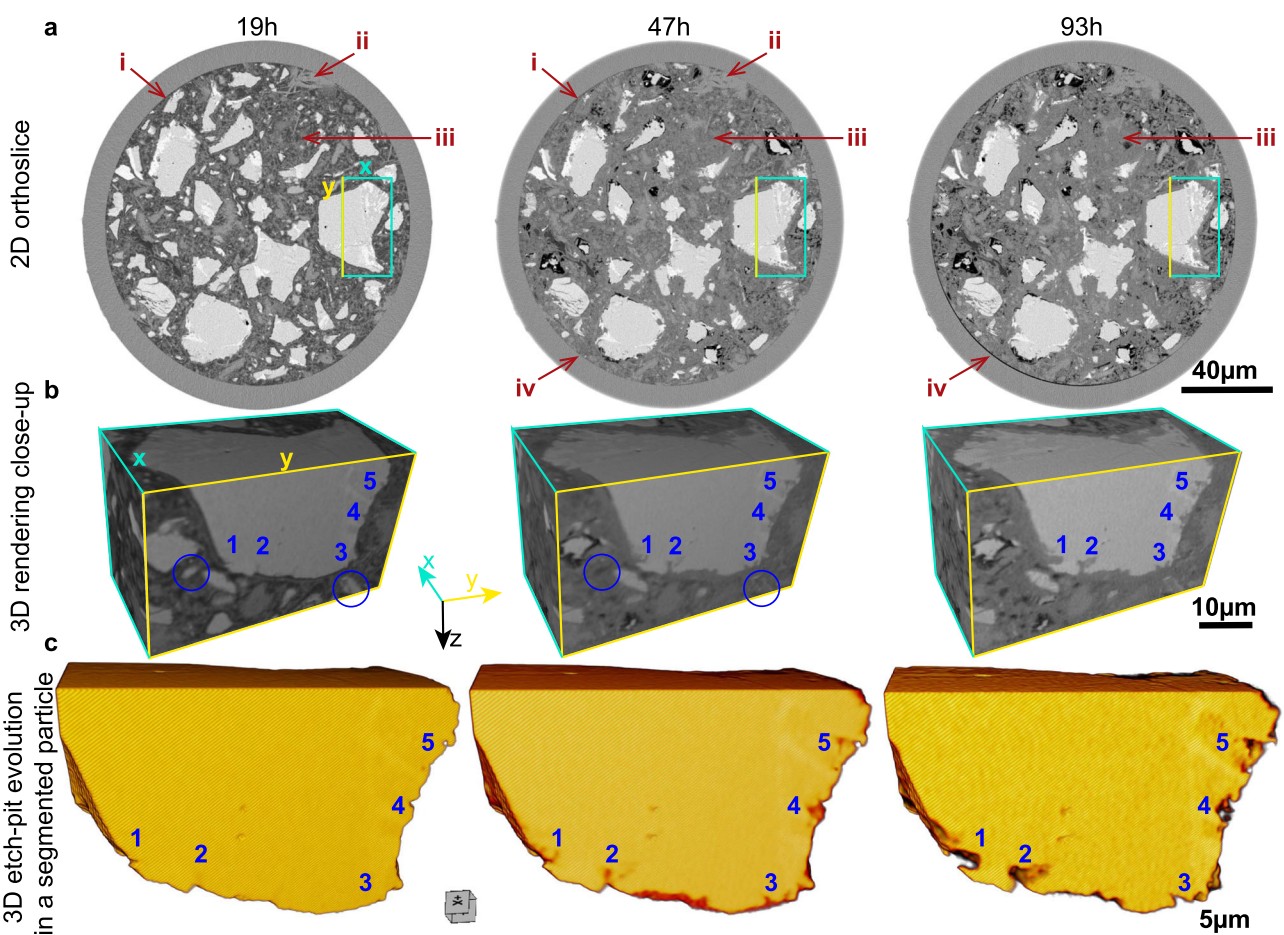

**Fig. 4 | 4D nanoimaging study highlighting the etch-pit evolution. a** PXCT orthoslices at the studied hydration ages showing the evolution of the PC-52.5-w/c~0.40 paste which includes: i) dissolution of cement particles, ii) portlandite growth, iii) C-S-H gel densification at 2 and 4 days, and iv) chemical shrinkage. Examples of these features are labelled with red arrows. **b** 3D rendering of a volume including a fraction of large alite particle highlighted in **a**, to show the evolution of five selected etch-pits, which are labelled. These images also show the full reaction of small alite particles, featured with blue circles. These 3D rendered views do not show exactly the electron densities as they are affected by visualization features like the lighting source. **c** 3D representation of the segmented particle, shown in the previous panels, to highlight the evolution of the etch-pits. It is noted that three branches within etch-pit #1 at 47 h coalesce at 93 h of hydration which may mean a reduction in surface with hydration time.

that the employed stoichiometry for C-S-H gel, i.e. $(CaO)_{1.80}(SiO_2)(H_2O)_{4.0}$[56,57], is an assumption and slightly smaller Ca/Si ratios, ~1.70, have also been reported[12,58]. On the other hand, ettringite and portlandite particles have larger sizes, usually ranging 1-5 $\mu m^3$, and they can be imaged in the present work. In any case, partial volume effect (the presence of components contributing below the spatial resolution of the measurements) is always taking place in cement pastes as some hydrates may have quite small particle sizes[17].

The crux of our results is the 4D submicrometer features of cement hydration, see Figs. 4 to 6. Supplementary Movie 1 also shows a summary of the main findings. It is underlined that PXCT readily distinguishes air and water porosities because of their difference in electron densities, 0 and 0.33 e·$Å^{-3}$, respectively; when the phase retrieval is carried out quantitatively[54]. The paste evolution is displayed in Fig. 4a, showing a partly reacted binder plenty of capillary water at 19 h. The main change from 19 to 47 h is the large consumption of capillary water (dark-grey regions in Fig. 4a-19h) and the densification of C-S-H gel. The main evolution from 47 to 93 h is the appearance of chemical shrinkage, evidenced by the development of many air-containing (black) regions. Importantly, Fig. 4c and Figs. S11–S12 show the evolution of etch-pits with time, including etch-pit coalescence, which contributes, in addition to the consumption of small alite particles, to the decrease of specific surfaces in the deceleration period.

The etch-pit growth rate was estimated, as detailed in SI, from the analysis of 27 dissolving regions in five alite grains. The resulting rate, between 19 and 47 h, was 41(29) nm/h. The etch-pit growth rate between 47 and 93 h was slower with a large variability, 7(8) nm/h, showing that the water diffusion is already limiting hydration. The large variability in the growth rates of the etch-pits is highlighted in Fig. S13. Etch-pits at very early ages, i.e. 2-4 h, were imaged by EM with sizes of ~15 nm[59] which is out of in situ PXCT capabilities.

Chiefly, the spatial dissolution rate of alite was determined from the study of the surface evolution of selected particles, see Fig. 5a and Figs. S14−S19, as examples. $C_2S$ particles were identified and excluded from this analysis, because they do not have C-S-H shells at 19 h of hydration. $C_3A$ particles were also recognised and discarded because of their smaller electron density values, i.e. ~0.91 e·$Å^{-3}$, 5% lower than that of alite. From 22 measurements along different surfaces, the dissolution rate between 19 and 47 h was 25(14) nm/h. This value compares well with 36 nm/h from the reaction zone model[8] but poorly with 84 nm/h obtained from the same dataset by using the boundary nucleation and growth model[60]. Moreover, this spatial dissolution rate can also be estimated from the segmentation results presented in the next subsection. Based on the quantitative analysis derived from Machine Learning (ML) segmentation of the PXCT datasets (for the $C_3S/C_2S$ class, dark brown colour code in the 3D visualization of Fig. 7),

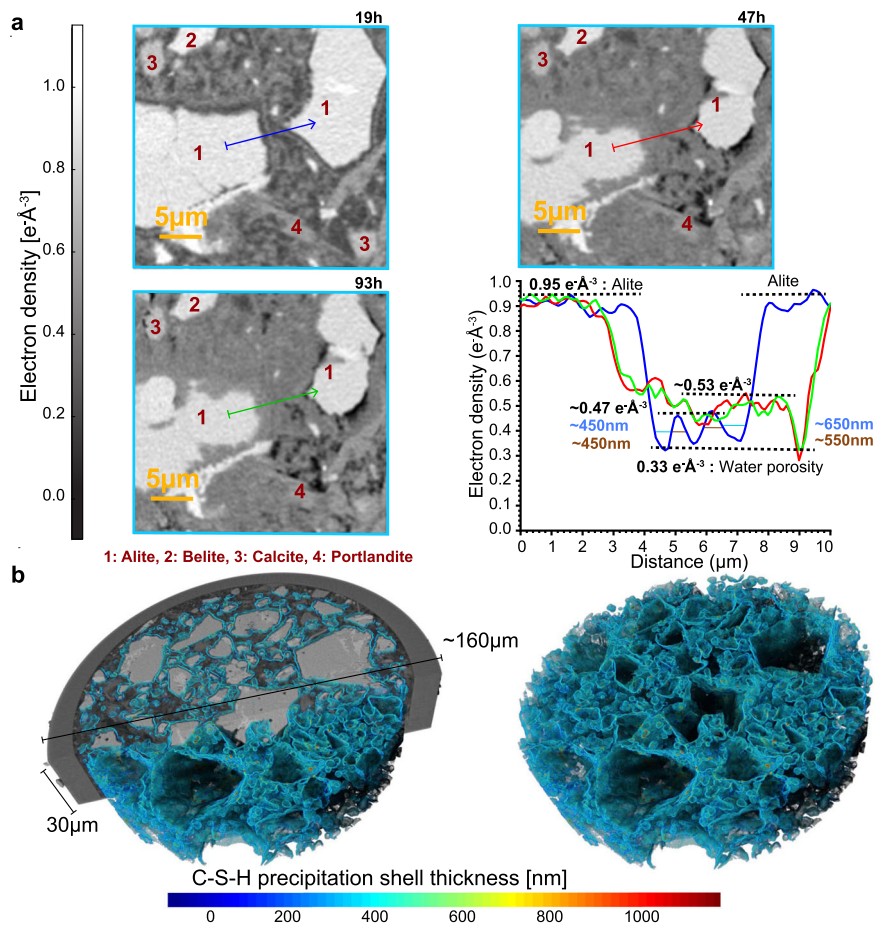

1: Alite, 2: Belite, 3: Calcite, 4: Portlandite

**Fig. 5 | C-S-H gel shells and their evolution. a** Selected 2D views of the PXCT data. The fourth panel displays the electron density profiles corresponding to the straight lines in the previous plots. At 19 h, the C-S-H shell covers every alite particle. The line profile at 19 h, blue trace in the fourth panel, shows water porosity (gap) between the alite particle and its shells. For these two particles, the shells have a thickness of ~550 nm and an electron density of ~0.47 e⁻Å⁻³. This C-S-H densifies at 2 days to ~0.53 e⁻Å⁻³. The alite particles partly dissolve and air porosity starts to develop (black regions). **b** 3D rendered views of the ML segmented C-S-H gel shells with the colour signalling its thickness for the 19 h tomogram. (left) View superimposing the C-S-H shells to half of the studied capillary. (Right) 3D view of the segmented C-S-H shells.

it is possible to derive an average spatial dissolution rate. Mathematical morphology tools were used to retrieve the outer layer of the segmented grains at 19 and 47 h. Subsequently, the average distance between these outer layers was computed for each grain, giving a mean value of 1.92 pixels (i.e. ~13 nm/h). This value is smaller than that obtained from the analysis performed in 2D slices for alite, 25 nm/h. However, it should be noted that in the segmentation calculation, alite and belite were classified together and therefore, the obtained spatial dissolution rate is underestimated as belite does not dissolve at early ages. Table S5, Rietveld analysis results, indicates that the amount of belite is half of that of alite during this stage. Therefore, the spatial dissolution rate can be corrected. The alite spatial dissolution rate should be close to 13/0.67 or 19 nm/h. This value agrees relatively well with 25 nm/h, obtained from 22 measurements in 2D slices, given the number of approximations which took place for both calculations.

The analysis of the 12 largest Hadley grains (hollow-shell microstructure, see Fig. 6a, b and larger view at Fig. S20)[14,15,61,62] gave 2.6(3) µm of size. Here, their time-evolution can be followed and it is noted that hollow regions are filled with water at 19 and 47 h but dried at 93 h, see Fig. 6a, directly evidencing the water diffusion through the C-S-H shells. Additionally, the analysis of the 15 smallest alite particles leaving an unhydrated, very small, core gave an average particle size of 3.4(5) µm. Therefore, it is concluded that alite particles smaller than ~3.0 µm are fully hydrated in the 4-19 h range, leading to a spatial dissolution rate of ~100 nm/h. The largest size of the Hadley grains found here

agrees well with previous works[61,62] (at 24 h) reporting a maximum size of 5.0 µm with shells of 500 nm. Hence, it seems to be a 3-4 fold difference between the spatial dissolution rate of small alite grains in the acceleration period and that of large alite grains in the deceleration stage.

Figure 5a and Figs. S14–S17 show views detailing the hydration of alite particles and showing the C-S-H gel shells that surround all alite grains enclosing a gap. Similar plots containing belite particles, Figs. S18–S19, did not show gaps, as expected. C-S-H shells on alite have been extensively analysed by EM[6,14,15,61,63], but here their electron density and spatial evolutions can be followed. The C-S-H shells for these two alite particles, Fig. 5a, have ~0.47 e⁻Å⁻³ which increases to ~0.53 e⁻Å⁻³ at 47 h. This means a very low mass density shell. For instance, ettringite, a phase with 32 crystallization water molecules has ~0.56 e⁻Å⁻³, and mature C-S-H gel, with $(CaO)_{1.8}(SiO_2)(H_2O)_{4.0}$ composition, i.e. including gel pore water, had 0.64 e⁻Å⁻³ [46]. As this result is critical for discarding diffusion as the mechanism for the deceleration period, a larger study was carried out. 20 shells were analysed giving 0.51(4) e⁻Å⁻³ and 500(120) nm, for the average electron density and thickness, respectively. Moreover, this study also yields the average width of the gap, 490(140) nm. At 24 h, the thickness of the shells and gaps were reported as ~500 and ~300 nm from EM[62]. Moreover, a gap of 490 nm developed in 15 h, time between the end of the induction period and the first nanoimaging measurement, which means an alite spatial dissolution rate of ~33 nm/h. Therefore, the dissolution rates of

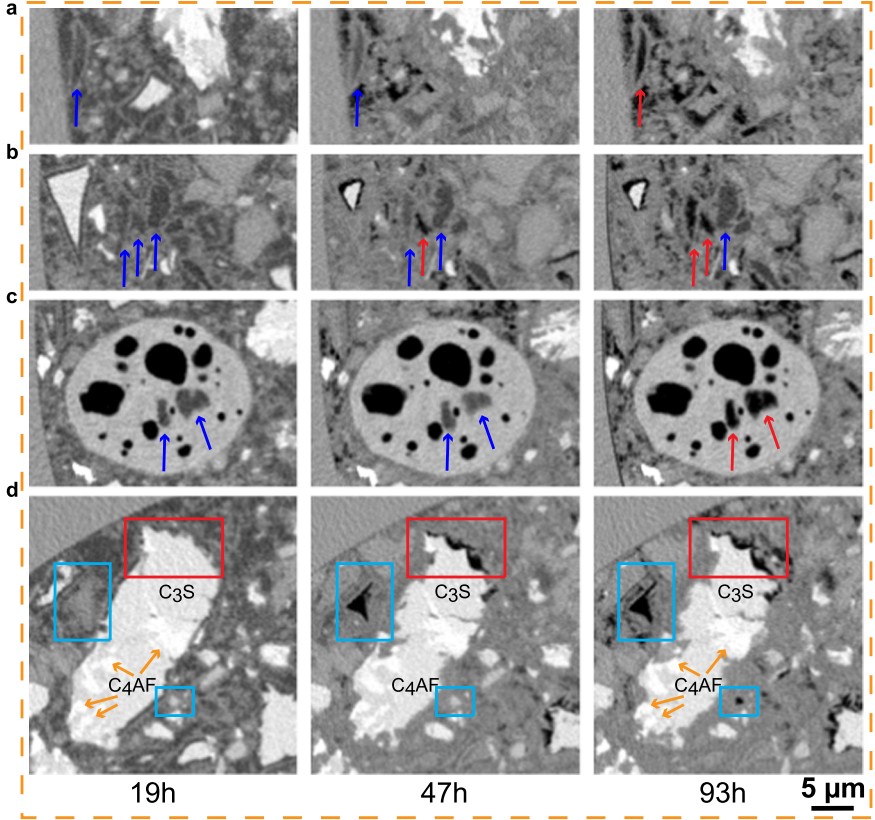

**Fig. 6 | Hydration time evolution of selected nano-features directly visualised by near field PXCT. a** Hollow-shell microstructure, also known as Hadley grain. The Hadley grains are fully hydrated small alite particles that contain a void within the original boundary of the anhydrous grain. The hollow regions are filled with water at 19 and 47 h (blue arrows) but dried at 93 h (red arrow) directly evidencing the porosity of the C-S-H shells. **b** Evolution of water porosity (dark-grey) to air porosity (black). It is noted that at 47 h, a tiny pore of about 1 μm size is already dried, but being very close to two larger water-filled pores, of sizes larger than 2 μm. This observation remarks the heterogeneity in cement hydration. It can be deduced that the relative humidity is not constant, at a given time, through the sample. This is due to a set of factors including the barriers to water diffusion and the crystallization/precipitation of different hydrates with quite different water contents, for instance ettringite and portlandite. **c** Evolution of water porosity inside a calcite grain, if connected to the surface. This calcite comes very likely from the limestone addition to the Portland cement as quantified in the anhydrous material, see Table S2. **d** Evolution of alite dissolution (hydration) which stops at the $C_4AF$ intergrown regions, highlighted by brown arrows. Moreover, alite hydration also stops as soon as air porosity (pore drying) develops on the surfaces of the anhydrous grains, see red rectangles. This panel also illustrates that (recently precipitated) hydrates can dissolve, see blue rectangles.

large and small alite grains differ at least three-fold, which should be considered for modelling. To further study the C-S-H shells, the components were segmented by ML as described in methods following the procedure summarized in Fig. S21. Subsequently, the C-S-H shells were segmented as detailed in Fig. S22. The results of the C-S-H shell segmentation are presented in Fig. 5b with a relatively constant shell size of ~450 nm. The Supplementary Movie 2 displays a full picture of the segmented shells. Fig. S23 shows a 2D comparison of the raw data and the resulting segmented shells yielding a reasonable good agreement.

Finally, Fig. 6 shows interesting (directly-observed) *nano-features*. Figure 6a, b exhibit the water diffusion through the C-S-H shells for some Hadley grains between 19 and 93 h of hydration. Larger regions are shown in Figs. S20 and S24, respectively. Figure 6c displays the water porosity evolution within a calcite particle evidencing that some small pore regions, 2-3 μm in size, were water filled at 19 and 47 h and dried at 93 h, see also Fig. S25. This observation directly explains the indirect result of water transport within limestone grains obtained by X-ray dark-field tomography[30]. Figure 6d, larger view in Fig. S26, shows how hydration progresses along the surface of an alite particle but it stops at $C_4AF$ intergrown regions, evidencing the importance of alkanolamines at early ages as accelerators[64]. Moreover, Fig. 6d also illustrates that C-S-H gel can dissolve to leave a dry capillary pore. Fig. S27 presents at 47 h the full

dissolution of a grain of a size of 4 μm at 19 h. This implies a very large dissolution rate, >75 nm/h, which is likely $C_3A$.

## Segmentations of the X-ray tomographies
Figure 7a displays orthoslices of the trained models overlaid on the three raw datasets and Fig. 7b shows the time evolution of the segmented components. For the PXCT study, the average electron densities and the segmented volumes are reported in Table S8. Moreover, Fig. S28 displays the evolution of the segmented water capillary porosity. It seems that at 19 h, the free water is preferentially located close to the walls of the capillary. This could be related to the 'wall effect' well known in mortars and concretes, where the cement paste content (water and fine cement particles) is slightly higher near the wall of the container respect to the larger aggregate particles which are preferentially arranged towards the centre. This feature, and its implications in the interfacial transition zone, has been extensively studied by numerous techniques, including synchrotron microtomography, see for example[65]. For cement pastes, higher porosity near the capillary wall has been observed by synchrotron microtomography[24]. For a water-rich alite paste, wall effect was clearly observed by PXCT where the resulting C-S-H gel had higher water content near the capillary wall[66]. In order to quantitatively study this feature, the scanned capillary was divided into two volumes, a central cylinder with half of the

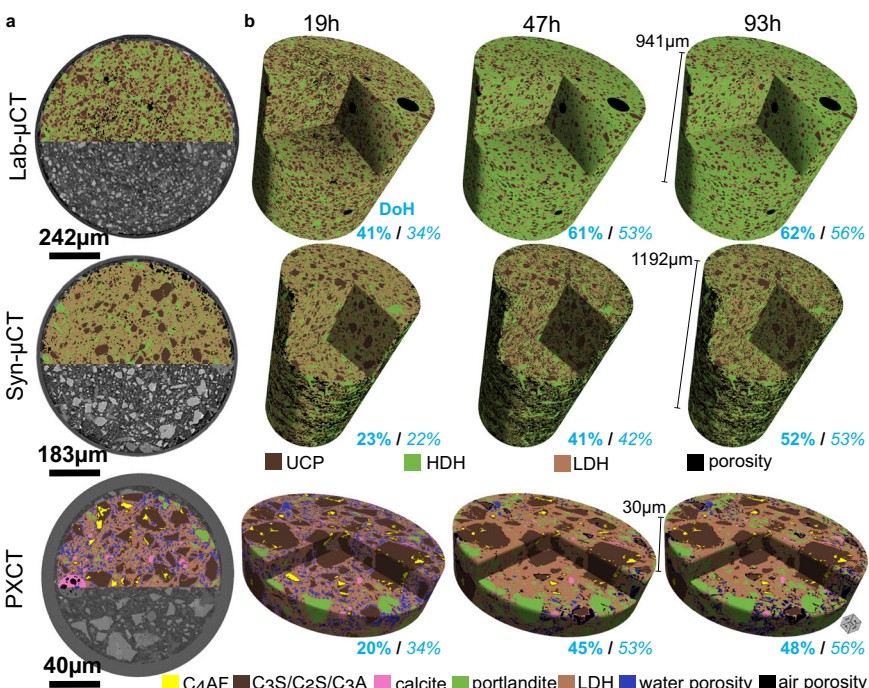

**Fig. 7 | ML training and segmentation results for the three datasets with different contrast mechanisms. a** ML trained models overlaid on the three raw datasets. **b** 3D rendering of the segmented volumes at the three studied hydration ages. The DoH values determined from microtomography (bold) are compared to the ones from calorimetry (italics). The number of quantified components in the Lab-μCT and Syn-μCT datasets are four: i) porosity (air and water), ii) LDH (low-density hydrates: mainly C-S-H gel and ettringite), iii) HDH (high-density hydrates:

mainly portlandite and calcite), and iv) UCP (anhydrous cement particles: all unreacted clinker phases). The number of quantified components in the PXCT datasets is seven: i) air porosity, ii) water porosity, iii) LDH (low-density hydrates: mainly C-S-H gel and ettringite), iv) portlandite, v) calcite, vi) $C_3A/C_3S/C_2S$, and vii) $C_4AF$. It is noted that with the quality of the data reported in this study (spatial resolution and electron density contrast), it is not possible to disentangle low-density from high-density C-S-H.

radius and the outer region up to the glass capillary wall. The mean electron densities were computed, but the voxels with electron density smaller than 0.24 e⁻Å⁻³, air porosity, were not included in order to minimise any bias due to the shrinkage/pore drying. The results for the centre volume were 0.614, 0.618 and 0.617 e⁻Å⁻³ for the 19, 47 and 94 h datasets, respectively. The corresponding mean electron density values for the outer region were 0.608, 0.610 and 0.601 e⁻Å⁻³. The 1% difference between the two regions at 19 h is quite small but not negligible. Hence, the degree of hydration could slightly be a function of the horizontal position of the particles.

The amount of anhydrous components, together with the initial values, allow determining the DoH from the three imaging studies which are reported in Table S9 and summarised in Fig. 7b. Table S9 also gives the DoH from calorimetry and LXRPD as references. The agreement for Syn-μCT is noteworthy likely due to the large volume probed and the relatively good spatial resolution. The DoHs from Lab-μCT are overestimated probably due to the relatively poor spatial resolution and contrast, which do not allow us to quantify the anhydrous particles smaller than ~3 μm. Finally, PXCT yields underestimated values for the DoH likely due to the limited scanned volume.

Importantly, the high resolution and excellent contrast of the PXCT study allowed us to track down the individual hydration evolution of 1407 particles with connected anhydrous volume, at 19 h, of 1 μm³ or larger. For this particle tracking statistical study, they were arbitrarily classified into four groups having connected volumes (μm³) of $1.0 \geq vol_1 > 27.0 \geq vol_2 > 216.0 \geq vol_3 > 1000.0 \geq vol_4$. A rough estimation of their sizes in μm could be $\sqrt[3]{(vol)}$ or $1.0 \geq size_1 > 3.0 \geq size_2 > 6.0 \geq size_3 > 10.0 \geq size_4$. The groups contained 1117, 204, 61 and 20 particles, respectively. The corresponding volume percentages with respect to the overall anhydrous cement particle volume at this hydration age, were 5.4, 12.9, 21.5 and 60.2%. It is not possible to calculate the DoH of these groups at 19 h because there are no reference

values at $t_0$. The degree of reaction, between 47 and 19 h, was 69.4%, 59.3%, 37.2% and 15.9% for the groups classified as $vol_1$, $vol_2$, $vol_3$ and $vol_4$, respectively, and they account for 12.9, 26.4, 27.7 and 33.0% of the total dissolved volume. In other words, the twenty particles of $vol_4$ group has a degree of reaction of 15.9% but it accounts for 33% of the dissolved volume in that period. A similar study can be carried out between 93 and 47 h. The degree of reaction during this period decreased to 14.7%, 11.6%, 6.7% and 3.6% for the corresponding groups. In terms of dissolved volume percentage in this period, the values are 6.8, 16.9, 25.3 and 51.0%, respectively. This simple analysis shows that 51% of the observed reaction during this diffusion-limited period is due to the 20 largest particles as they account for a very large fraction of the volume. The strong decrease of hydration rate in this time range is in line with a diffusion controlled hydration stage.

Finally, Fig. 8a displays the hydration evolution of the segmented anhydrous cement particles. It is readily visible that smaller particles dissolve faster than larger ones. Moreover, the segmentation output allows us to classify the particles and to follow their volumes and cumulative volumes, see Fig. 8b. The cumulative volume results for Lab-μCT and PXCT are very similar and in agreement with the expected DoH from calorimetry showing a large variation between 19 and 47 h and a small variation at 93 h. Moreover, the hydration of PC-42.5 is slower because of their larger average particle sizes, with similar DoH variation between the two studied time intervals, see Fig. 8b, in full agreement with their calorimetric traces, see Fig. 1b. This reflects a good accuracy of the obtained results. Finally, it should be noted that the scanned length in the vertical direction, 30 μm, for PXCT is limited taking into account that some alite grains with sizes of 20 μm, or slightly larger, are present in PC cements. This was mitigated by imaging 160 μm in the transversal direction. This type of experiment will benefit from imaging cylindrical volumes with 60-100 μm of height. However, with the current experimental procedure, this would lead to

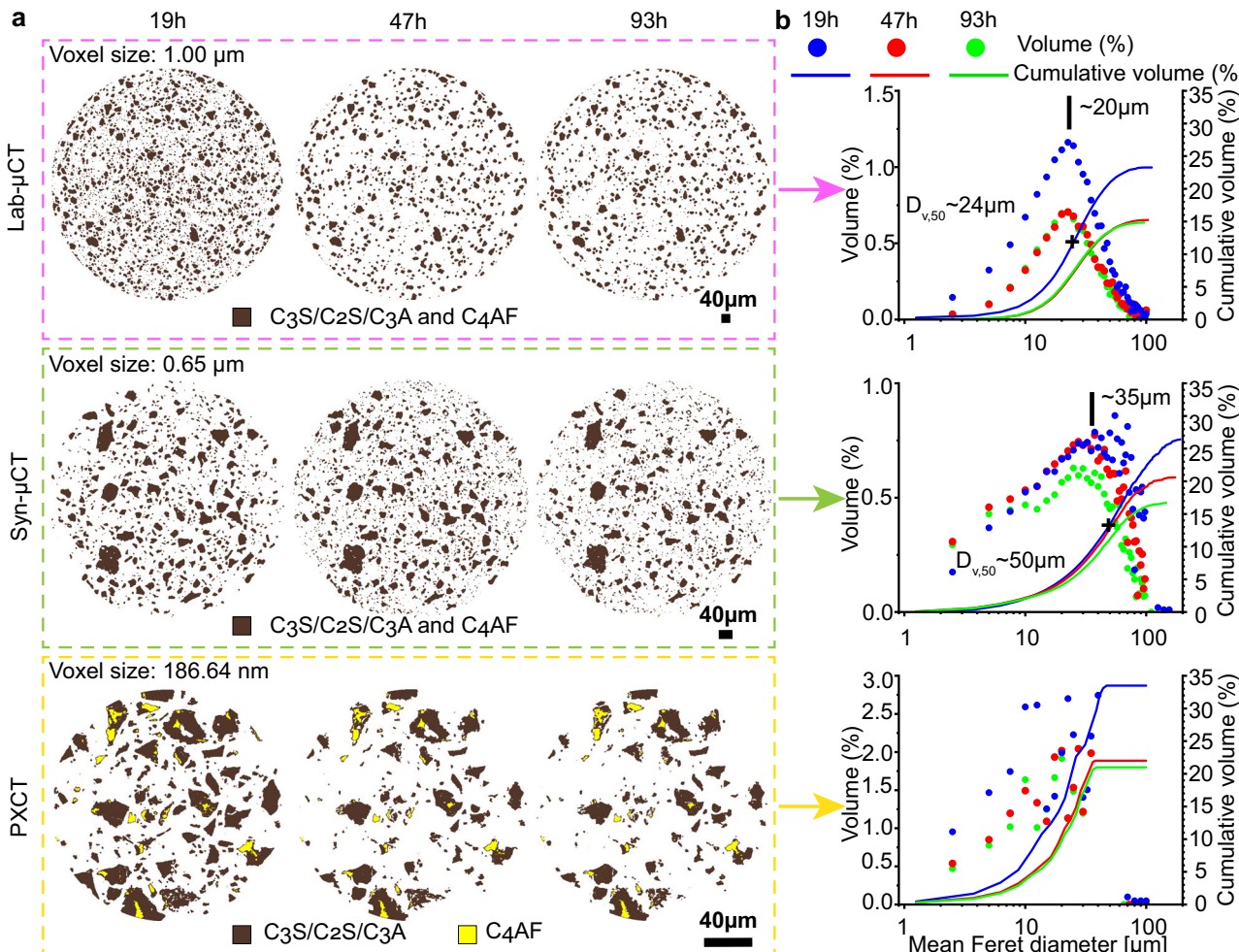

**Fig. 8 | Hydration study as a function of time and cement particle sizes. a** 2D views of the segmented anhydrous cement particles as a function of time. PXCT data shows that $C_4AF$ hydrates little up to 93 h. This is likely due to the low w/c ratio in the scanned volume and its slow hydration rate. **b** Volumes and cumulative volumes for the anhydrous cement particles as a function of the particle size, represented in logarithmic scale for easy comparison with Fig. 1. The particle sizes are computed as the mean Feret diameter frequency which measures object size along directions. The particles are represented grouped in sets of 2.5 microns, i.e. 0-2.5, 2.5-5.0 μm, etc. The maximum and $D_{v,50}$ values for the Lab-μCT and Syn-μCT are given in the panels. These data are scattered for PXCT because the limited height of the studied cylinder yields a poor representative elementary volume for this feature.

acquisition times larger than 7-9 h and therefore changes due to hydration could take place during an acquisition. Procedures for faster data collection are being explored and they are discussed below.

The hydration of PC, in relevant conditions, has been measured with very high spatial resolution, i.e. 250 nm, and contrast. As expected but not directly measured so far, the hydration of Portland cement at 1 day or earlier is dominated by the small particles, smaller than 3 μm, and the hydration after 3 days is very much dependent of the large alite particles, larger than about 10 μm. The nanoimaging work shows the C-S-H gel shells surrounding every alite grain which does not preclude diffusion. The measured alite spatial dissolution rates, ~100 nm/h for small grains in the acceleration period and ~25 nm/h for large particles in the deceleration stage impose constraints on the cement hydration models. The alite etch-pit growth rate between 19 and 47 h has been estimated as ~40 nm/h, which decreases to ~7 nm/h in the 47 to 93 h interval. Moreover, etch-pit coalescence, the merging of different branches, has also been visually observed. However, better spatial resolution is required for a thorough etch-pit growth rate quantification. The configuration employed here already allows studying the roles of admixtures (accelerators, retarders, superplasticizers, etc.) by measuring C-S-H shell density, C-S-H gel spatial distribution and alite spatial dissolution and etch-pit growth rates. For instance, it would be

possible to measure if the acceleration produced by $CaCl_2$ is due to a lower density of the C-S-H shells (higher water diffusion) or if it is mainly due to higher calcium supersaturation.

The current spatial resolution of in situ near-field PXCT, i.e. ~250 nm, could be improved by increasing the number of projections, without larger acquisition times. Moreover, so far the time resolution is modest, i.e. 3-4 h for a complete tomogram, but this is expected to improve in fourth-generation synchrotron sources with tailored beamlines for ptychography. Time resolutions of ~1 h will open the way to accurately study the processes in the acceleration period. However, in these cases, radiation damage could be an issue if the total dose is not kept low. Higher time resolution does not necessarily imply higher doses and therefore possibly larger radiation damage. For instance, sparsity techniques could be coupled to PXCT in order to decrease the overall acquisition time for the whole series by as much as 90%, as recently reported[67]. Another approach could be to use machine learning/deep learning for denoising of datasets collected with much less X-ray dose[68]. On the other hand, gamma irradiation of Portland pastes, mortars and concretes is known to produce water radiolysis finally leading to $H_2$ microbubbles and calcium peroxide, $CaO_2 \cdot 8H_2O$[69]. Therefore, the signatures of these species should be monitored for studies with high X-ray doses. Higher spatial resolution

and shorter acquisition times will allow to thoroughly study the acceleration stage which will be beneficial to rationally design new accelerator admixtures with the final aim of developing low carbon cements with competitive mechanical strengths and durability performances. Moreover, dissolution-precipitation processes with moderate reaction rates take place also in several other fields like geochemistry or biomaterials, which could benefit from the reported investigation.

## Methods

### Material provenance and initial characterization

Two different types of commercial cement were used: a CEM I 52.5 R (PC-52.5) and a CEM I 42.5 R (PC-42.5) which conform to EN 197−1. The full characterization of these anhydrous materials is performed with X-ray Fluorescence (XRF) and Laboratory X-ray Powder Diffraction (LXRPD) using the Rietveld Method, see Fig. S1 and Tables S1, S2. Moreover, a polycarboxylate ether (PCE) superplasticiser, containing 35 wt% of the active matter, was employed to efficiently fill the very narrow capillaries required for the PXCT nanoimaging study. The characterisation of this PCE has been recently reported[70].

### Textural analysis

The textural characterization was carried out in dry conditions. Particle size distribution (PSD) data were measured by laser diffraction in a MasterSizer 3000 equipment (Malvern). Specific surface areas were determined by $N_2$ adsorption isotherms in ASAP 2420 equipment (Micromeritics, USA). The air permeabilities were measured with the Blaine fineness apparatus (Controls) according to EN 196−6. The density of the samples was measured with a helium Pycnometer (Accupyc II 1320 Pycnometer, Micromeritics). The resulting data are reported in Table S3.

### Isothermal calorimetry

The pastes were prepared using the PC-52.5 and PC-42.5 and with w/c mass ratios of 0.40 and 0.50, respectively. For PC-52.5, 0.43 wt% (by weight of cement) of PCE was used. Water and superplasticizer were mixed by magnetic stirring for 1 min. Then, the cement was mixed with the water/suspension and shaken, for 1 min manually and 1 min with a laboratory vortex mixer. Finally, the pastes were introduced into the glass ampoules. The calorimetric data were taken in an eight-channel Thermal Activity Monitor (TAM) instrument. Data were collected for up to 7 days at 25 °C and at 20 °C and for PC-52.5 and PC-42.5, respectively. The first 45 min were required for the thermal stabilization of the system. The employed conditions mimic the ones used for the imaging studies.

### Laboratory X-ray Powder Diffraction (LXRPD) and Data Analysis

The same paste prepared for the calorimetry measurement (PC-52.5-w/c = 0.40) was used to fill the capillaries of ~1 mm in diameter. LXRPD measurements were collected on a D8 ADVANCE diffractometer (Bruker AXS) using strictly monochromatic Mo-K$\alpha_1$ radiation ($\lambda = 0.7093$ Å). This diffractometer is located at SCAI, University of Malaga. The incident beam was formed by a primary monochromator with a focusing mirror and a 2 mm anti-scatter slit. Moreover, 2.5° Soller slits were used for the incident and transmitted beams. An EIGER detector (from DECTRIS, Baden, Switzerland) was used which is optimised for Mo anodes. This was used with an aperture of 4 × 21 degrees, working in VDO mode. Data collection was performed from 3 to 35° (2θ) for 2 h and 10 min. Rietveld quantitative phase analysis was performed with GSAS software.

### Laboratory X-ray computed microtomography experiment (Lab-μCT)

Lab-μCT experiments were carried out at ~25 °C for the same capillary used in the LXRPD data collection and scanning the same region with

time. Lab-μCT experiment was performed on a SKYSCAN 2214 (Bruker) scanner at SCAI, University of Malaga. Images were obtained using an X-ray tube with a LaB$_6$ source filament and employing a 0.25 mm Al foil to minimise the beam hardening effect. This source was operated at 55 kV and 130 μA. The CCD3 detector with a physical pixel size of 17.427 μm was set in a middle position with a source-to-detector distance of 315.449 mm and a source-to-sample distance of 9.051 mm which yielded a voxel size of 1.00 μm (binning 2 × 2). Finally, the projections were acquired every 0.22° over 360° with a total of 1637 projections per tomogram and using an exposure time of 1.9 s. This results in an overall recording time of 3.5 h per dataset. Image reconstruction of the CTs was carried out using Bruker NRecon software (version 2.1.0.1) and by applying Gaussian smoothing and beam hardening correction.

Because Lab-μCT and LXRPD data were acquired in the same capillary, but not in the same equipment, the description of the timing is important. Lab-μCT data were collected from 17.5 h until 21.0 h, using cement-water mixing time as the reference. This dataset is labelled as 19 h. Then, the capillary was transfer to the diffractometer, in the same room, and the LXRPD data were taken from 21 h 15 min to 23 h 25 min. This dataset is labelled as 22 h. The data collections at 47 h and 93 h followed the same protocol but it is less important at these hydration ages, as the kinetics are slower.

### Synchrotron X-ray computed microtomography experiment (Syn-μCT)

PC-42.5 with a w/c ratio of 0.50 was used to prepare the paste for this experiment and PCE was not required. The paste was manually mixed for 3.5 min and then introduced in a capillary of ~0.7 mm of diameter. Inline propagation-based phase-contrast microtomographic data were acquired at ID19 beamline of the European Synchrotron (ESRF) in Grenoble, France. The measurements were performed at 21.5 °C, temperature of the experimental hutch, using a photon energy of 19 keV. The distance between the sample and the detector was 15 mm. The total time to record a full tomogram was 6 min with 0.05 s exposure time. 6000 projection angles were acquired over a 360 degree tomographic scan. For reconstructions, Paganin phase retrieval of the projections was performed[51]. The resulting voxel size was 0.65 μm. Further experimental details are given in the SI.

### Near-field ptychographic X-ray computed tomography (PXCT)

The paste employed for the in situ nanoimaging study was PC-52.5 with a nominal w/c ratio of 0.50 and 0.43 wt% of PCE. The suspension was mixed with a mechanical stirrer at 800 rpm for 3.5 min and then introduced in a glass capillary of 200 μm of nominal diameter. Near-field PXCT data were taken at the cSAXS beamline of the Swiss Light Source (SLS) at the Paul Scherrer Institute (PSI), Villigen, Switzerland. Near-field ptychography[47] is a variant of X-ray ptychography[42,44] in which the sample is scanned across a coherent divergent illumination and magnified images of the sample are recorded on the detector at each scanning position. Ptychographic phase retrieval algorithms are employed to reconstruct the complex transmission function of the specimen, with both absorption and phase, at each angular projection. We then repeat measurements of many projections at different incident angles of the X-rays onto the sample and combine them using standard tomographic reconstruction methods to obtain 3D maps of the electron density and the absorption coefficient of the specimen. In near-field ptychography, if a sufficient number of projections are recorded, the spatial resolution is limited by the magnified pixel size, which is determined by the pixel size of the detector, the distance between the sample and the detector, the divergence of the illumination and the position of the specimen from the point source of the beam, i.e. the focus. We performed our measurements using a high-stability instrument designed for high-resolution PXCT working in air[71,72] and at the hutch temperature of 25 °C, using a photon energy of

8.93 keV. The thickness of the capillary in the imaged region was 160 μm. Therefore, it was scanned with a FOV of 186 μm in order to have more than 10 μm of air outside the capillary, which is required for quantitative phase imaging. The total time to record a full tomogram was between 3 and 4 h, including dead time during the motion of stages in between acquisitions. The voxel size was 186.64 nm. Further experimental details are given in the SI.

## Tomographic data analysis

A supervised Machine-Learning (ML) image analysis approach was used to segment the different components of imaged samples, using the IPSDK Explorer software (version 3.2.0.0 for Windows™, Reactiv'IP, Grenoble, France). The plot profiles along the scans electron density/ grey value slices and the 3D rendering visualization were done by using Dragonfly software (version 2022.1 for Windows™, Object Research Systems (ORS) Inc., Montreal, Canada). More information about data analysis can be found in the SI.

## Data availability

The twelve reconstructed tomograms 'raw' data in tiff format, and the laboratory characterisation data, have been deposited and they can be freely accessed on Zenodo at https://doi.org/10.5281/zenodo.7030107, and used under the Creative Commons Attribution license.

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

## Acknowledgements

Financial support from PID2019-104378RJ-I00 research grant, which is co-funded by FEDER, is gratefully acknowledged. SLS is thanked for granting beamtime at cSAXS under proposal 20210147. ESRF is thanked for beamtime at ID19. ToScA (United Kingdom) is gratefully acknowledged for awarding Jim Elliott Award to Shiva Shirani, which supported her stay at ESRF. Dr. Manuel Guizar-Sicairos is thanked for his assistance with the ptychography data processing. I.R.S. is thankful for funding from PTA2019-017513–I.

## Author contributions

M.A.G.A. conceived, designed and supervised this study. S.S. and I.S. did initial rheological and laboratory-tomographic studies to fill the 200-micron capillaries. S.S. and A.M.-C. carried out the laboratory characterization. M.A.G.A., S.S., A.C., A.D, P.T., M.H. applied for beamtime at the SLS and designed the PXCT experiment. S.S., A.D., P.T and M.H. carried out the synchrotron ptychographic experiment. S.S., B.L. and A.R. conducted the synchrotron microtomographic experiment. I.R.S. did the laboratory diffraction and microtomographic experiments. S.S. did all the X-ray imaging data analysis with assistance of M.A.G.A., A.D. and A.C. The machine learning segmentation was carried out by S.S. under E.B.'s supervision. M.A.G.A. wrote the first draft. S.S. prepared all the figures, with help of A.C. for bivariate and Rietveld plots. All authors discussed the results and commented on the manuscript.

## Competing interests

The authors declare no competing interests.
