## [Peer Review File · Nature Communications]

4D nanoimaging of early age cement hydrationREVIEWER COMMENTS

Reviewer #1 (Remarks to the Author):

The strength of this study, in my opinion, is in the exploration of a new nanoimaging technique, PXCT, to study the early hydration of Portland cement. With the unprecedented spatial resolution and contrast of PXCT, in situ identification of the evolution of various mineral phases, the C-S-H gel shells, etch pits, water pores and air pores, etc. became possible, hence the corresponding cement dissolution and precipitation processes at early ages associated with them was studied, and qualitative and quantitative results were obtained. The results are of potential interest to researchers working in X-ray computerized tomography and could illustrate a new characterization direction for the cement-based materials community.

The approach, data analysis and interpretation are valid, comprehensive and correct, and the evidence presented justify the conclusions. The literature is adequately cited. The clarity and accessibility of the text is good, and the results have been provided with sufficient context and consideration of previous work.

There are several issues however that I think should be clarified to help improve the work.

1. In the manuscript, it is said "In situ near-field data were taken as detailed in methods..... radiation damage cannot be discarded but it is small, if any. (Page 5) " "Time resolutions of ~1 h will open the way to accurately study the processes in the acceleration period. However, in these cases, radiation damage could be an issue if the total dose is not kept low. (Page 12)" Will radiation damage affect the visualization results of the evolution of water porosity towards air porosity with time? In addition, migration of C-S-H gel was observed. Is it possible that this was also affected by radiation damage? How will the effect of radiation damage be considered in the future research with higher time resolution?

2. The evolution of water porosity towards air porosity with time was observed around some Alite grains. How will this affect further hydration of Alite and etch pits growth when there is no water around?

3. In the manuscript, it is said "The hollow regions of the Hadley grains are filled with water at 19 and 47 h but dried at 93 h, see enlarged pictures to the right. This illustrates that most of the capillary pores with sizes larger than ~1 μm are already water emptied at 93 h of hydration, see bottom right. (Figure S19)", "The enlarged views (bottom) show the evolution of porosity within the paste, where several pores of sizes smaller than ~2 μm are dried (red arrows) at 93 h but other larger, pores keep filled with water (blue arrows). (Figure S23)" Is there a contradiction between those two conclusions, for the pores to be empty at 93 h, the former is $>1\mu\text{m}$ and the latter is $<2\mu\text{m}$? Does the water migrate randomly?

4. The water to cement ratio is relatively higher near the capillary wall than that in the center. In Figure S27, wall effect in the 19h capillary tube sample could be observed. Will the degree of hydration of cement particles of the same size be affected by wall effect at different locations? It should be addressed in the manuscript.

Reviewer #2 (Remarks to the Author):

In this manuscript, the authors report the study of the early stage of cement hydration using mainly near-field ptychographic X-ray computed tomography. They provide a unique combination of spatial & temporal resolution + field of view. It allows the determination of quantitative values for dissolution rates and etch-pit growth rates. A comparison with data obtained by lab and synchrotron X-ray micro-tomography is also proposed. The paper shows an extended collection of data using state of the art technique and an interesting 3D data analysis based on machine learning process. Nevertheless, the manuscript shows some incoherencies and approximations, the results are not sufficiently described and discussed, especially the ones reported in the abstract.

Thus, I cannot support this work for publication in Nature communications in its present form. Some specific comments and remarks linked to my previous general comment:
For the PXCT study, the size of the capillary used is different between the different sections and figures 160 μm vs 200 μm . This information is important. Indeed, it is highlighted (in the supplementary methods) that the sample should be smaller than the FOV (186 μm) in order to have quantitative results ("The field of view must be larger than the size of the capillary to include an air region at both sides of the sample, which is needed for successful tomographic reconstructions and for quantitative contrast.").
It is mentioned in the first paragraph of the "result and discussion" section that the particles size of the sample PC-52.5 for PXCT study are finer than the sample described in the table S3 and used for the lab measurements (XRD, tomography...). How is it done? What is the impact on the comparison of the results between the different measurements?
The w/c ratio value for the sample used for PXCT measurements differs from 0.4 to 0.5 when described in the SI file, in situ multicontrast X-ray tomographic studies of cement hydration and method sections.
What are the units in Table2 ?

In figure 2:

- In b, it can be seen that the images of the Lab-uCT scan are more blurred at 93h than at 19h. Why? Why does the analysis performed in Figure S5 not show the same tendency?
- On the contrary, Figure S7 clearly shows a degradation on the spatial resolution that is not observed by eye on the images.
- It is concluded that the resolution is estimated at two pixels (How?). It means that the resolution of the PXCT study is 374 nm. Then, how representative and trustable are the values given in figure S13 -Figure S15 ?
- Looking at the figure c: Can you link the difference of the grey-value in the diagram of lab-uCT and syn-uCT to the different behaviour of the two different cements ?
- There is a comparison between the lab-uCT diagram and the PXCT one. What is the abscissa axis label for the lab-uCT ? I understood that it was grey-value. If yes, what is the link to compare grey-values of absorption tomography to electron density obtain by PXCT ?
- Is the diagram of figure 2c the same than Figure S10? Are they representative of the same sub-volume ? Is "the largest possible volumes without including the glass capillary walls" is equivalent to the $\sim 1 \times 10^5 \mu\text{m}^3$ VOI mentioned in the "tomographic data analysis section" of SI for the PXCT study?

Looking at the spatial resolution analysis:

How many interfaces have been studied as in figures S2-3-4 to determine the resolution estimations? What is the standard deviation? Can the calculation linked to the figure s5-s6-s7 be detailed? How the resolution values are deduced from the curves?

Are the volumes used to determine the data of Table S7 and Figure S10 equal to the VOI?

It can be seen that there is a difference between the theoretical electron density of the components and the measured one. How do you explain the difference? Why did you choose to use the theoretical value to indicate the different components in figure S10 ? Why is there no measured values for air, water and LDH ? Which electron densities have been used to train the machine learning for those phases?

Why the grey values of the 3D rendering in figure 4 are different from one-step to another?

How has been the etch pit growth rate evaluated?

At the beginning of page 6 it is mentioned the spatial dissolution rate of alite. Since alite, C3A and belite has been classified in the very same category for the segmentation, how is it possible to make the difference between them when looking at C-S-H gel shells. I guess that the belite particles are not surrounded by gel shells because the DoH is equal to zero in table S5 but what about the differentiation between C3S and C3A?

Do we have an idea of how many particles of alite $< 3\mu\text{m}$ were present in the sample at the beginning of the experiment?

Is the figure 5b representative of the 19h step?

At the end of the page 9 and in figure 8, it is mentioned, "PXCT yields underestimated values for the DoH likely due to the limited scanned volume" and "These data are scattered for PXCT because the limited height of the studied cylinder yields a poor representative elementary volume for this feature". It is in contradiction with the conclusion given in page 4 "giving confidence to the relevancy of the nanoimaging results in spite of the limited amount of volume scanned to have submicrometer resolution". Can you give more info and comment on the representativeness of the volume scanned by PXCT?

What is the UCP volume mentioned in page 10?

How is it possible to link the results obtained from 2D figures 5 and S13 to S18 to the ones obtained with the 3D segmentation? Are the results giving the same spatial dissolution rate?

Can you specify what you mean by quantification in this sentence: "Etch-pit growth rate, ~40 nm/h, and coalescence have also been measured but better spatial resolution is required for etch-pit quantification"?

At the end of the "implications and outlook" paragraph it is written that "The current spatial resolution of in situ near-field PXCT, ~370 nm, can be improved by increasing the number of projections, without larger acquisition times" which, from my knowledge, is false. While in the supplementary info section it is reported and especially pointed out that "The resolution obtained, see subsection dedicated to the spatial resolution, was limited by the number of projections, which was chosen to have reasonable scan times".

Is the sample perfectly stable from one measurement to another? Is there a need to "realign" the volume from one-step to another? Is there a need for volume registration? If yes, how has it be done ?

Reviewer #3 (Remarks to the Author):

Major revision is needed.

This paper presents a method to observe the hydration process of cement using 4D nanoimaging. The idea is novel, but there are some problems:

[1]The literature should be updated, more literature should be in recent three years.

[2]In the Introduction, the disadvantages of the references should be summarized clearly to emphasize the importance of this work. The difficulties of 4D nanoimaging also need to be summarized.

[3]In Fig.7, the time selection is very strange."19h","47h","93h". what is the reason. The title of this paper focuses on the early hydration. why not select some early hydration points? for example, Non-contact multiple-frequencies AC impedance instrument for cement hydration based on a high-frequency weak current sensor.

[4]This paper is interesting, cement pastes are multi-phase materials. Some phases are overlapped with others in some localized regions. Some nano-scale C-S-H and CH may be mixed together. It is said that the intrinsic size of C-S-H is about 5 nm(doi.org/10.3390/fractalfract5020047). Therefore, in Fig.4 nanoimaging, C-S-H may not be identified in the scale bar 5 μm . Therefore, at least, the typical size of each phase during the cement hydration should be listed out from literature, so that reader can compare comprehensively.

[5]As we know, some commercial PCs have some mineral admixtures. What about the XRF results of these two commercial PCs in this work.

[6]Fig. 3. Left figure is terrible. We can not identify which is which. It is only black and white.

[7]From Fig.7, the samples used in PXCT have serious carbonation. However, other samples do not have such carbonation. Besides, in Fig.7, how do authors identify the different kinds of C-S-H gels. Is it from AFM test according to porosity or NMR according to silicate calcium chain(doi.org/10.1016/j.conbuildmat.2020.118807 & doi.org/10.1016/j.measurement.2021.110019)

[8]In Fig.6,authors claim they can monitor the evolution of water porosity (dark-grey) to air porosity (black). Does it mean that water can be distinguished by near field PXCT? As I know, only neutron can examine the presence of water.

[9]In Fig.8,PXCT test results seem that C4AF content does not change much. It is not reasonable. Common sense tells us that the hydration rate is $C3A > C3S > C4AF > C2S$.

Reviewer #4 (Remarks to the Author):

A superbly presented, detailed study of cement hydration using various X-ray methods. I would recommend a review by someone expert in the cement hydration field, since I cannot comment on the impact of the results from the point of view of the application. What I can say is that the ptychographic imaging is a significant real-world demonstration of the near-field approach and, to my mind, represents the current state-of-the-art.

I have no issue recommending the article for publication, subject only to minor improvement of the English - which although always comprehensible could do with a proof-read.

Response to reviewers document. Reference NCOMMS-22-47460-T

Title. “4D nanoimaging of early age cement hydration”

To help the readability and understanding of this revision, our answers are highlighted **in green** in the response to reviewer’s letter and the changes are highlighted **in blue** in the revised version of the manuscript and in the S.I. file.

Reviewer #1 (Remarks to the Author):

The strength of this study, in my opinion, is in the exploration of a new nanoimaging technique, PXCT, to study the early hydration of Portland cement. With the unprecedented spatial resolution and contrast of PXCT, in situ identification of the evolution of various mineral phases, the C-S-H gel shells, etch pits, water pores and air pores, etc. became possible, hence the corresponding cement dissolution and precipitation processes at early ages associated with them was studied, and qualitative and quantitative results were obtained. The results are of potential interest to researchers working in X-ray computerized tomography and could illustrate a new characterization direction for the cement-based materials community.

The approach, data analysis and interpretation are valid, comprehensive and correct, and the evidence presented justify the conclusions. The literature is adequately cited. The clarity and accessibility of the text is good, and the results have been provided with sufficient context and consideration of previous work.

Author reply: Many thanks for acknowledging the main contributions of this work.

There are several issues however that I think should be clarified to help improve the work.

1.1. In the manuscript, it is said “In situ near-field data were taken as detailed in methods..... radiation damage cannot be discarded but it is small, if any. (Page 5)” “Time resolutions of ~1 h will open the way to accurately study the processes in the acceleration period. However, in these cases, radiation damage could be an issue if the total dose is not kept low. (Page 12)”

Will radiation damage affect the visualization results of the evolution of water porosity towards air porosity with time? In addition, migration of C-S-H gel was observed. Is it possible that this was also affected by radiation damage? How will the effect of radiation damage be considered in the future research with higher time resolution?

1.1. Author general reply: With the data currently available, we can only speculate on these interesting questions. It is known that the radiolysis of water in cement pastes yields, in a first stage, electrons and H₂O₂. After a cascade of reactions, the identified products (under gamma irradiation) are H₂ and calcium peroxide, i.e. CaO₂·8H₂O [see for instance, Bouniol and Aspart, Cement and Concrete Research (1998) “Disappearance of oxygen in concrete under irradiation: the role of peroxides in radiolysis”]. There are many papers dealing with gamma-irradiation of mortars and concretes but we are not aware of any investigation dealing with the radiation damage mechanism by ‘mild’ X-ray irradiation, that it does not necessarily need to be identical.

1.1. Author reply-a: We do not think that possible radiation damage will significantly modify the water porosity towards air porosity evolution. However, at this stage, this is a speculation that we prefer not to include in the revised version of the manuscript.

1.1. Author reply-b: C-S-H migration (dissolution and re-precipitation) has been observed in some small volumes/regions of the sample and we think this is due to local under/over saturation fluctuations rather than radiation damage. Unfortunately, we do not have data to firmly establish this point. More high-resolution imaging studies are required.

1.1. Author reply-c: Higher time resolution does not necessarily imply higher doses (and therefore larger radiation damage). For instance, sparsity techniques can be coupled to PXCT in order to decrease the overall acquisition time for the whole series by 90% [Gao et al., *Sci. Adv.* 7, eabf6971 (2021)]. Another approach could be to use ML for the reconstruction and denoising of datasets collected with (much) less X-ray dose [Hendriksen, et al., *Sci. Rep.* 11, 11895 (2021)].

The two key important points are now addressed in the revised version on page 14: “... Higher time resolution does not necessarily imply higher doses and therefore possibly larger radiation damage. For instance, sparsity techniques could be coupled to PXCT in order to decrease the overall acquisition time for the whole series by as much as 90%, as recently reported⁶⁴. Another approach could be to use machine learning/deep learning for denoising of datasets collected with much less X-ray dose⁶⁵. On the other hand, gamma irradiation of Portland pastes, mortars and concretes is known to produce water radiolysis finally leading to H₂ microbubbles and

calcium peroxide, $\text{CaO}_2 \cdot 8\text{H}_2\text{O}$ ⁶⁶. Therefore, the signatures of these species should be monitored for studies with high X-ray doses....” Three new references have been added.

1.2. The evolution of water porosity towards air porosity with time was observed around some Alite grains. How will this affect further hydration of Alite and etch pits growth when there is no water around?

1.2. Author reply: As soon as there is no contact with water, alite hydration stops. This can be seen in Figs. 5 and 6 and also in some figures in the S.I. However, with the data at hand, it is not possible to know if later-age water diffusion could re-wet some alite grain, and hence, hydration could progress. This is the reason we prefer not to speculate too much about this. In any case, this feature is now explicitly stated in the manuscript in the caption of Fig. 6 by adding “... Moreover, alite hydration also stops as soon as air porosity (pore drying) develops on the surfaces of the anhydrous grains, see red rectangles...”. Figure 6 has been consequently updated with the corresponding red rectangles.

1.3. In the manuscript, it is said “The hollow regions of the Hadley grains are filled with water at 19 and 47 h but dried at 93 h, see enlarged pictures to the right. This illustrates that most of the capillary pores with sizes larger than $\sim 1 \mu\text{m}$ are already water emptied at 93 h of hydration, see bottom right. (Figure S19)”, “The enlarged views (bottom) show the evolution of porosity within the paste, where several pores of sizes smaller than $\sim 2 \mu\text{m}$ are dried (red arrows) at 93 h but other larger, pores keep filled with water (blue arrows). (Figure S23)”

Is there a contradiction between those two conclusions, for the pores to be empty at 93 h, the former is $>1 \mu\text{m}$ and the latter is $<2 \mu\text{m}$? Does the water migrate randomly?

1.3. Author reply-a: We do not think that there is a contradiction between the highlighted observations. Our interpretation is the importance of heterogeneity, which is key in cement hydration when analysed with enough spatial resolution and contrast. Just in Fig. 6, panel b, it can be directly seen that at 47 h, a pore of about $1 \mu\text{m}$ size is dried, but being very close to two larger water-filled pores, of sizes larger than $2 \mu\text{m}$. At first sight, this seems to contradict Kelvin-Laplace equation that relates the relative humidity (RH) to the size of the dried pores. This implies that the pores are emptied of water from largest to smallest sizes as hydration progresses. This is indeed correct in the absence of impediments for water diffusion (which is not the case as C-S-H gel is a barrier to diffusion) and if the RH is constant within the paste, which is also not held. For instance, in regions where ettringite crystallizes, consuming a lot of water, the RH drops more than in other regions where only C-S-H/portlandite precipitates, which have lower water contents. This has been clarified in the caption of figure 6, b-panel, by adding the following statements. “... It is noted that at 47 h, a tiny pore of about $1 \mu\text{m}$ size is already dried, but being very close to two larger water-filled pores, of sizes larger than $2 \mu\text{m}$. This observation remarks the heterogeneity in cement hydration. It can be deduced that the relative humidity is not constant, at a given time, through the sample. This is due to a set of factors including the barriers to water diffusion and the crystallization/precipitation of different hydrates with quite different water contents, for instance ettringite and portlandite...”

1.3. Author reply-b: We do not have enough data to model/deal with the migration of water. This is indeed a very interesting issue that will be treated when we collect a major number of tomograms at different ages.

1.4. The water to cement ratio is relatively higher near the capillary wall than that in the center. In Figure S27, wall effect in the 19h capillary tube sample could be observed. Will the degree of hydration of cement particles of the same size be affected by wall effect at different locations? It should be addressed in the manuscript.

1.4. Author reply: This is a quite important observation that was not treated in the submitted version. The ‘wall effect’ –the increased cement paste content near the wall of the container respect to the large aggregate particles in mortars and concretes– is well known (in mortars and concretes) and it has been extensively studied by numerous techniques including electron microscopy and synchrotron microtomography. The ‘wall effect’ in pastes, where water is partly segregated towards the wall of the capillary, has also been reported but it has been less studied. We have addressed the reviewer’s comment by adding the following set of sentences on page 11, and the corresponding two new references. “... It seems that at 19 h, the free water is preferentially located close to the walls of the capillary. This could be related to the ‘wall effect’ well known in mortars and concretes, where the cement paste content is slightly higher near the wall of the container respect to the larger aggregate particles which are preferentially arranged towards the centre. This feature, and its implications in the interfacial transition zone, has been extensively studied by numerous techniques, including synchrotron microtomography, see for example⁶². For cement pastes, higher porosity near the capillary wall has been observed by synchrotron microtomography.²⁴ For a water-rich alite paste, wall effect was clearly observed by PXCT where the resulting C-S-H gel had higher water content near the capillary wall.⁶³ In order to quantitatively study this feature, the scanned capillary was divided into two volumes, a central cylinder with half of the radius and the outer region up to the glass capillary wall. The mean electron densities were computed, but the voxels with electron density smaller than $0.24 \text{ e} \cdot \text{\AA}^{-3}$, air porosity, were not included in order to minimise any bias due to the shrinkage / pore

drying. The results for the centre volume were 0.614, 0.618 and 0.617 $\text{e}^{-\text{\AA}^{-3}}$ for the 19, 47 and 94 h datasets, respectively. The corresponding mean electron density values for the outer region were 0.608, 0.610 and 0.601 $\text{e}^{-\text{\AA}^{-3}}$. The 1 % difference between the two regions at 19 h is quite small but not negligible. Hence, the degree of hydration could slightly be a function of the horizontal position of the particles....”

Two new references have been added:

(62) *Energy Procedia* **114**, 5109–5117 (2017)

(63) *Langmuir* **31**, 3779–3783 (2015)

Reviewer #2 (Remarks to the Author):

In this manuscript, the authors report the study of the early stage of cement hydration using mainly near-field ptychographic X-ray computed tomography. They provide a unique combination of spatial & temporal resolution + field of view. It allows the determination of quantitative values for dissolution rates and etch-pit growth rates. A comparison with data obtained by lab and synchrotron X-ray micro-tomography is also proposed. The paper shows an extended collection of data using state of the art technique and an interesting 3D data analysis based on machine learning process. Nevertheless, the manuscript shows some incoherencies and approximations, the results are not sufficiently described and discussed, especially the ones reported in the abstract. Thus, I cannot support this work for publication in Nature communications in its present form.

Author reply: thanks for your general comment which explicitly acknowledges the key contribution of this work “... They provide a unique combination of spatial & temporal resolution + field of view. ...”.

Some specific comments and remarks linked to my previous general comment:

2.1. For the PXCT study, the size of the capillary used is different between the different sections and figures 160 μm vs 200 μm . This information is important. Indeed, it is highlighted (in the supplementary methods) that the sample should be smaller than the FOV (186 μm) in order to have quantitative results (“ The field of view must be larger than the size of the capillary to include an air region at both sides of the sample, which is needed for successful tomographic reconstructions and for quantitative contrast.”).

2.1. Author reply: We thank the reviewer for allowing us to clarify this issue, which is very important. The glass capillary has a nominal diameter of 200 μm , however it is not fully uniform in its length. In the imaged vertical region, it has a thickness of 160 μm . Therefore, it was scanned with a FoV of 186 μm in order to have more than 10 μm of air outside the capillary, which is needed as a reference for quantitative phase imaging. This is now explicitly stated in the method section of the main manuscript. On page 15: “... introduced in a glass capillary of 200 μm of nominal diameter...” and “... The thickness of the capillary in the imaged region was 160 μm . Therefore, it was scanned with a FOV of 186 μm in order to have more than 10 μm of air outside the capillary, which is required for quantitative phase imaging...”.

2.2. It is mentioned in the first paragraph of the “result and discussion” section that the particles size of the sample PC-52.5 for PXCT study are finer than the sample described in the table S3 and used for the lab measurements (XRD, tomography...). How is it done? What is the impact on the comparison of the results between the different measurements?

2.2. Author reply: We now are aware that our wording was misleading. We have used two PCs, one PC-52.5 and another PC-42.5. PC-52.5 was used for the PXCT study, and for the laboratory characterization: μ -CT, Rietveld quantitative phase analysis and calorimetry. The PC-52.5 employed in these measurements was identical. Indeed, PC-42.5, employed for the synchrotron μ -CT, has slightly larger particle sizes as measured in Fig. 1a, Table S3 and Fig. 8. This is clarified in the revised version by explicitly stating on page 3: “PC-52.5 was used for the PXCT and laboratory μ -CT imaging studies and the additional laboratory characterization. PC-42.5, with slightly larger average particle size, was used for the synchrotron μ -CT imaging study.”

2.3. The w/c ratio value for the sample used for PXCT measurements differs from 0.4 to 0.5 when described in the SI file, in situ multicontrast X-ray tomographic studies of cement hydration and method sections.

2.3. Author reply: The nominal w/c ratio employed for the paste, that filled the capillary in the PXCT study, was 0.50. However, it is very challenging to control the w/c homogeneity within very thin capillaries. Therefore, the w/c ratio of the scanned volume in the PXCT study was measured and the value was slightly smaller. This is now fully clarified by adding on page 4 “... The nominal w/c mass ratio employed to fill the PXCT narrow capillary, 200 μm of nominal diameter, was 0.50, see methods. However, it is very difficult to accurately control the w/c ratio in very thin capillaries. Therefore, the w/c ratio of the scanned volume for the PXCT measurement was measured as previously published⁴⁶ and detailed in a subsection of the S.I. The w/c ratio of the scanned volume in the PXCT study was 0.41.”.

2.4. What are the units in Table 2?

2.4. Author reply: The units are wt%, and this is corrected in the heading of Table S2.

2.5. In figure 2: - In b, it can be seen that the images of the Lab- μ CT scan are more blurred at 93h than at 19h. Why? Why does the analysis performed in Figure S5 not show the same tendency?

2.5. Author reply: The FSC for laboratory microtomography, shown in Fig. S5, does not indicate higher degradation of the spatial resolution at 93 h. This is a direct observation. Fig 2b (laboratory data) shows a change which is not blurring but the consequence of hydration. At 19 h, the image has more free water and anhydrous components, hence it has higher contrast. At 93 h, the consumption of water to give hydrated phases results in a poorer contrast between them and the unhydrated phases, which may appear as a blurring effect.

2.6. On the contrary, Figure S7 clearly shows a degradation on the spatial resolution that is not observed by eye on the images.

2.6. Author reply: We thank the reviewer for allowing us to clarify this issue. The spatial resolution, as measured by FSC, for the 47 and 93 h tomograms, i.e. 470-500 nm, is slightly poorer than that of the 19 h, 430 nm. This was very likely due to the employed scanning step size that was 6 μ m for the 19 h tomogram (resulting in 3h 55 min of total acquisition time), and 7 μ m for the other two tomograms (resulting in 3h 6 min of total acquisition time). This was detailed in the experimental section, but now we understand that this was not enough. Therefore, this is now clarified in Figure S7 caption by adding “The slightly better spatial resolution measured for the 19 h tomogram is very likely due to the smaller scanning step size, i.e. 6 μ m, and the corresponding larger acquisition time, i.e. 3h 55 min. The scanning step size for the other two tomograms was 7 μ m, yielding 3h 6 min of acquisition time.”

2.7. It is concluded that the resolution is estimated at two pixels (How?). It means that the resolution of the PXCT study is 374 nm. Then, how representative and trustable are the values given in figure S13 –Figure S15?

2.7. Author reply-a: We thank him/her for allowing us to clarify this point about the spatial resolution which is indeed very important. Be aware that this point is intimately linked to the point 2.11, below. We acknowledge that this point was not treated with the required depth in the submitted version. We have carried out further studies to characterise this key feature: Now in the revised version “... The spatial resolution was characterised by two approaches as recently reported⁵². The procedures are thoroughly detailed in Supplementary Information (SI). On the one hand, the spatial resolution can be determined by the edge sharpness across selected interfaces. A point spread function (PSF) used to determine the spatial resolution of the images as ISO/TS 24597 defines the Gaussian radius of the PSF as the resolution, which equals a change between 25 %–75 % grey value along the studied interfaces.⁸⁸ The spatial resolutions, determined by this approach, were 250(25) nm, 264(25) nm, 272(34) nm, 748(19) nm and 2.21(17) μ m, for PXCT-19h, PXCT-47h, PXCT-93h, Syn- μ CT and Lab- μ CT datasets, respectively. As examples of this procedure, Figures S2-S4 display line profiles of sharp interfaces between high (i.e. alite) and low density (i.e. porosity) components within the capillaries. On the other hand, Fourier-shell-correlation (FSC)⁵³ has also been employed to estimate spatial resolution. Figures S5-S7 displays the FSC traces for the three imaging modalities. The agreement between both approaches is satisfactory for Syn- μ CT and Lab- μ CT, but not for PXCT. The worse spatial resolution estimated by FSC for PXCT is very likely due to the low number of projections, i.e. 420, which make the subtomograms employed in the FSC calculation severely undersampled.” Reference 52 has been added: (52) *Nat. Nanotechnol.* **15**, 356–360 (2020).

2.7. Author reply-b: Moreover, we have included a full new subsection in S.I. which reads:

“Spatial resolution analysis.

The spatial resolution was characterised by two approaches as recently reported¹⁷. On the one hand, it can be determined from the grey-value changes in line profiles through the edge sharpness of the interfaces. A point spread function (PSF) used to determine the spatial resolution of the images as ISO/TS 24597 defines the Gaussian radius of the PSF as the resolution, which equals to a change between 25 %–75 % grey value along the studied interfaces.¹⁸ Here, a common interface present in the three imaging modalities has been selected for the line profiles: the glass capillary wall – air (i.e. exterior of the capillaries). We have measured 25 interfaces in every tomogram, which allowed us to determine the average spatial resolution and its associated standard deviation. Moreover, as examples, Figures S2-S4 display line profiles of sharp interfaces between high (i.e. alite) and low density (i.e. porosity) components within the capillaries. The spatial resolutions, determined by this approach, were 250(25) nm, 264(25) nm, 272(34) nm, 748(19) nm and 2.21(17) μ m, for PXCT-19h, PXCT-47h, PXCT-93h, Syn- μ CT and Lab- μ CT datasets, respectively.

On the other hand, FSC plots⁵ have been also computed. The traces are displayed in Figures S5-S7 giving spatial resolution values of 430 nm, 470 nm, 500 nm, 650 nm and 1.9 μ m, for PXCT-19h, PXCT-47h, PXCT-93h, Syn- μ CT and Lab- μ CT datasets, respectively. Moreover, the FSC trace for PXCT-19h shows a smooth decrease in the 0.0-0.2 spatial frequency range, which is likely due to the hydration of cement during the 4-hour measurement. As expected, this behaviour is not shown at later ages.

It should be noted that the agreement between the spatial resolution results between the edge sharpness approach and FSC method is satisfactory for Syn- μ CT (750 vs. 650 nm) and Lab- μ CT (2.2 vs. 1.9 μ m) datasets. However, the agreement between these two approaches is not good for PXCT (for instance, 250 vs 430 nm at 19 h). The poorer resolution estimated by FSC can be explained because the angular sampling is very tight, i.e. 420 projections, so the two employed subtomograms in the FSC, each of 210 projections, were significantly undersampled compared to the number of voxels across the diameter of the sample. This means that the correlation between two such undersampled tomograms can give a low estimation of the spatial resolution. This feature is not observed for Syn- μ CT and Lab- μ CT as the total number of projections were 6000 and 1637, respectively. In other words, the subtomograms with half the number of projections were not undersampled for these two imaging modalities.”

2.7. Author reply-c: In our opinion, we can trust the results shown in Figures S13-S15 as these features have dimensions 400-500 nm, significantly larger than the spatial resolution of PXCT study, which is determined as \sim 250 nm, see just above.

2.8. Looking at the figure 2c: Can you link the difference of the grey-value in the diagram of lab-uCT and syn-uCT to the different behaviour of the two different cements?

2.8. Author reply: It is not straightforward to link the different behaviour of PC-52.5 in Lab- μ CT and PC-42.5 in Syn- μ CT in Figure 2c because the Paganin phase retrieval is highly sensitive to the initial porosity. However, the expected behaviour: faster reactivity of PC-52.5 at early ages than PC-42.5, because of the finer particle sizes, is beautifully shown in Figure 8 b. As dictated by the particle sizes and as measured by calorimetry, PC-52.5 reacts faster between 19 and 47 h and then very little from 47 to 93 h, see Fig. 1b panel. This is clearly shown in the cumulative volume traces from Lab- μ CT, see Figure 8 (top). Conversely, from calorimetry, PC-42.5 reacts slower between 19 and 47 h but it keeps reacting between 47 to 93 h, see Fig. 1b panel. This is evident in the cumulative volume traces obtained from Syn- μ CT, see Figure 8 (intermediate). This was discussed in the segmentation subsection of the manuscript.

2.9 There is a comparison between the lab-uCT diagram and the PXCT one. What is the abscissa axis label for the lab-uCT? I understood that it was grey-value. If yes, what is the link to compare grey-values of absorption tomography to electron density obtain by PXCT?

2.9. Author reply-a: Yes, the abscissa axis label for lab- μ CT is “grey-value”. This is corrected in the revised version of the manuscript.

2.9. Author reply-b: It is not possible to quantitatively link the grey-values in the lab- μ CT and the electron density in PXCT. The electron density values in PXCT are related to the imaginary part of the refractive index of every component. This relationship is quantitative but some nuances should be taken into account, such as partial volume effects and that amorphous phases and solid solutions have (slightly) variable compositions and hence, electron density values. Conversely, the grey-values in Lab- μ CT are related to the attenuation coefficient of every component (i.e. the real part of the refractive index). Moreover, the relationship between grey-values and the attenuations is not quantitative because the polychromatic nature of the laboratory radiation. This has been clarified in the revised version by adding on pages 4-5“... The grey-values in the Lab- μ CT study, see Fig. 2c (top panel), are related to the attenuation coefficients of the components in this PC-52.5 paste, but the relationship is not quantitative due to the polychromatic nature of the employed radiation. Conversely, the electron density values obtained for the same paste by PXCT are quantitative. Therefore, the grey scales in the Lab-CT and the electron densities in the PXCT datasets cannot be directly related as they derived from the imaginary and the real part of the refractive index of every component.”.

2.10. Is the diagram of figure 2c the same than Figure S10? Are they representative of the same sub-volume? Is “the largest possible volumes without including the glass capillary walls” is equivalent to the $\sim 1 \times 10^5 \mu\text{m}^3$ VOI mentioned in the “tomographic data analysis section” of SI for the PXCT study?

2.10. Author reply: Yes, the VOI were exactly the same and as big as possible without incorporating the glass capillary walls. There was a problem with the colours of the different traces in Fig S10, to be consistent with the main text, and this is now corrected. The revised figure S10 has the 93 h trace in green.

2.11. Looking at the spatial resolution analysis: How many interfaces have been studied as in figures S2-3-4 to determine the resolution estimations? What is the standard deviation? Can the calculation linked to the figure s5-s6-s7 be detailed? How the resolution values are deduced from the curves?

2.11. Author reply: For the reported number, we have studied 25 interfaces for every type of imaging modality. The average values and the associated standard deviations are now given, see point 2.7 above. The calculations are fully detailed now in the S.I.

2.12. Are the volumes used to determine the data of Table S7 and Figure S10 equal to the VOI?

2.12. Author reply: We thank the reviewer for allowing us to clarify this. The VOI used to obtain Figure S10 is the biggest possible volume leaving out the capillary. However, for Tables S7 and S8, the measured electron densities for the different phases/components are carried out for the large available particles. This was explicitly stated at the bottom of Table S8 but not in Table S7, which was misleading. This has been corrected by adding the information at the bottom of Table S7. “# Electron densities, from particle picking, were obtained by the average of 10 cubes for the capillary; 5, 4, 5 and 6 grains for portlandite, calcium carbonate, alite and belite, respectively.”

2.13. It can be seen that there is a difference between the theoretical electron density of the components and the measured one. How do you explain the difference? Why did you choose to use the theoretical value to indicate the different components in figure S10 ? Why is there no measured values for air, water and LDH ? Which electron densities have been used to train the machine learning for those phases?

2.13. Author reply-a: We thank the reviewer for allowing us to clarify this point which was not dealt with in the original submission. The differences between the theoretical electron densities and the measured ones are mainly due to partial volume effects. This is now explicitly stated in the revised version on pages 6-7 “... The differences between the theoretical electron densities and the measured ones are mainly due to partial volume effects. For instance, portlandite, i.e. $\text{Ca}(\text{OH})_2$, has a theoretical electron density value of $0.69 \text{ e} \cdot \text{\AA}^{-3}$. The measured values at 19 and 93 h were $0.62(2)$ and $0.651(5) \text{ e} \cdot \text{\AA}^{-3}$, see Table S7. These numbers are 6-10% smaller than the theoretical one, with the difference being higher than the errors of the measurements, which are estimated in 2-3%^{46,54}. This difference is very likely due to the presence of residual water porosity below the spatial resolution of the measurements, which we refer to partial volume effects.” Reference 54 has been added: (54) *Phys. Rev. B* **85**, 020104 (2012).

2.13. Author reply-b: We prefer to indicate the theoretical values of electron densities in Fig. S10 because they are free of partial volume effects and they can be used in future works with higher spatial resolution and therefore, where the implications of the partial volume effects will be smaller.

2.13. Author reply-c: Concerning the measured values for air, water and LDH. They are now included in Table S7. “... Moreover, 5 cubes at 19 h gave the reported measured electron density for capillary water. Similarly, 5 cubes at 93 h were computed to obtain the value for air. Finally, the electron density of LDH (low density hydrates) was measured at 93 h in 5 cubes yielding $0.56(1) \text{ e} \cdot \text{\AA}^{-3}$ that it corresponds to ettringite and/or C-S-H as they cannot be distinguished.”

2.13. Author reply-d: Concerning the electron density values used to train the machine learning. It should be noted that we did not use a single value of electron density but a range of approximately 5% of the measured electron density value for the initial classification and training of the different components. Afterwards, based on the obtained results, some voxels were relabelled and further training was carried out based on the electron density values and on the morphology of the components. We acknowledge that this information was not given and this is now corrected in the revised version of the S.I. by adding “... The initial classification was based on the electron densities with a variation of ~5% of the measured values, from selected volumes, which are given in Table S7.”

2.14. Why the grey values of the 3D rendering in figure 4 are different from one-step to another?

2.14. Author reply: The change in grey values between each step in Fig 4b is caused by the “diffuse light” effect used for 3D rendering. We have to note that the 3D rendered views do not show exactly the grey values (or electron densities) but they are affected by the visualization features. This is now clarified in the figure caption. “... These 3D rendered views do not show exactly the electron densities as they are affected by visualization features like the lighting source.”

2.15. How has been the etch pit growth rate evaluated?

2.15. Author reply: We acknowledge that the procedure for etch pit growth rate estimation/evaluation was not detailed and this is now corrected two-fold. We have added a small subsection in the S.I. detailing the procedure and its limitations. Furthermore, the results have been updated in the main text. This point is also very much related to the point 2.22, from this reviewer, see below. It is noted that this is not a full quantitative determination but an estimation, as the spatial resolution of our imaging study, ~250 nm, is not good enough for a thorough study of these tiny features.

Now in the S.I.: **Etch pit growth rate evaluation.**

The estimation of the etch-pit growth rate was based on the analysis of 27 etch-pits from 5 different large alite grains. It is noted that the etch-pits have irregular 3D shapes and therefore, for its spatial dissolution rate estimation, some simplifications were undertaken. Moreover, the spatial resolution of this PXCT work, ~250 nm, is limited for accurate analyses. Therefore, we consider this approach as an estimation. Firstly, etch pits were visually selected from grains with sizes larger than 10 μm . Secondly, their overall shapes were compared in two

hydrating steps. Then, two envelopes from pixels with at least 90% of the electron density of alite were developed. The estimated/calculated distance (in pixels) was computed between these edges for the deepest perpendicular length. These values were converted to dissolution rate by taking the ratio with respect to the time between measurements. The result for the analysis between 19 and 47 h datasets gave 6.1 pixels of average distance which is equivalent to 41(29) nm/h. There was large variability in the rates, the fastest being 110 nm/h and the slowest being 10 nm/h. From this investigation, it is not possible to know if this large variability comes from the heterogeneity in the defects within these regions, or if other variables like the spatial resolution of this work and the local water-to-cement ratio variations are also playing important roles. More imaging studies are necessary to establish this. The very same 27 etch-pits were also analysed between 47 and 93 h datasets. In this case, the etch-pit growth rate was slower 7 nm/h, showing that the water diffusion is already limiting hydration at four days.

Now, in the manuscript on page 7: "... The etch-pit growth rate was estimated, as detailed in S.I., from the analysis of 27 dissolving regions in five alite grains. The resulting rate, between 19 and 47 h, was 41(29) nm/h. The etch-pit growth rate between 47 and 93 h was slower with a large variability, 7(8) nm/h, showing that the water diffusion is already limiting hydration."

2.16. At the beginning of page 6 it is mentioned the spatial dissolution rate of alite. Since alite, C3A and belite has been classified in the very same category for the segmentation, how is it possible to make the difference between them when looking at C-S-H gel shells. I guess that the belite particles are not surrounded by gel shells because the DoH is equal to zero in table S5 but what about the differentiation between C3S and C3A?

2.16. Author reply: This comment is quite related to 2.21 below, both related to the C3S spatial dissolution rate. Here we clarify that, in the original submission, the spatial dissolution rate of alite was not determined from the segmentations but from the analysis of 22 surfaces of particles of different sizes. Firstly, C₂S particles were excluded because they do not have a C-S-H shell at 19 h of hydration. C₃A particles were also discarded as they have a smaller electron density, $\sim 0.90 \text{ e}^{-\text{\AA}^{-3}}$ instead of the $\sim 0.95 \text{ e}^{-\text{\AA}^{-3}}$ of alite particles. This is now clarified in the text by "Chiefly, the spatial dissolution rate of alite was determined from the study of the surface evolution of selected particles, see Fig. 5a and Figs. S13-S18, as examples. C₂S particles were identified and excluded from this analysis, because they do not have C-S-H shells at 19 h of hydration. C₃A particles were also recognised and discarded because of their smaller electron density values, i.e. $\sim 0.91 \text{ e}^{-\text{\AA}^{-3}}$, 5 % lower than that of alite. From 22 measurements along different surfaces, the dissolution rate between 19 and 47 h was 25(14) nm/h..."

2.17. Do we have an idea of how many particles of alite <3µm were present in the sample at the beginning of the experiment?

2.17. Author reply: From the particle size distribution measurement by laser diffraction, we can estimate that ~20 vol% of PC has sizes smaller than 3 µm. Because the cement has 61 % of alite, the initial cement had about 12 vol% of alite with these particle sizes. Moreover, the scanned region of the capillary is filled with a paste containing a w/c mass ratio of 0.40. Due to the large difference in densities (water=1 g/cc and cement=3.12 g/cc) the capillary contained at the time of mixing, 56.1 vol% of water and 43.9 vol% of cement particles. Therefore, the alite volume at the beginning of the experiment with particle sizes smaller than 3 µm is estimated as ~5.3 vol%. As the scanned volume of paste was $\sim 5 \cdot 10^5 \mu\text{m}^3$ (the total volume was $8.1 \cdot 10^5 \mu\text{m}^3$ but this included the glass capillary and the air) and it contained about $\sim 26000 \mu\text{m}^3$ of alite particles smaller than 3 µm. A back of the envelope calculation indicates that if all particles have 3 µm, and they have cubic shape, it should be around 1000 particles. Conversely, if all particles have 1 µm isotropic size, it should be around 26,000 particles. This coarse calculation agrees reasonably well with the segmented number of alite particles at 19 h of hydration with sizes between 1 and 3 µm, i.e. 1117 particles which come from partly hydrated larger ones, stated in the manuscript (given the number of approximations, and taken into account that all small particles are dissolved at early hydration ages). At this stage, we prefer not to include this calculation in the manuscript as all data to carry out this coarse estimation are in the paper. However, if the reviewer thinks differently, we could include this calculation in the supplementary information file.

2.18. Is the figure 5b representative of the 19h step?

2.18. Author reply: Yes, fully representative. We forgot to include that description in the figure caption. This mistake has been corrected by adding "... for the 19 h tomogram."

2.19. At the end of the page 9 and in figure 8, it is mentioned, "PXCT yields underestimated values for the DoH likely due to the limited scanned volume" and "These data are scattered for PXCT because the limited height of the studied cylinder yields a poor representative elementary volume for this feature". It is in contradiction with the conclusion given in page 4 "giving confidence to the relevancy of the nanoimaging results in spite of the limited amount of volume scanned to have submicrometer resolution". Can you give more info and comment on the representativeness of the volume scanned by PXCT?

2.19. Author reply: We do not think that there is a contradiction between these statements. In fact, it is highlighted from the very beginning "... in spite of the limited amount of volume scanned to have submicrometer resolution" We have also carried out Syn- μ CT and Lab- μ CT, because the scanned volume in PXCT is limited. It is explicitly stated in the experimental section, and visually in Figure 7, that PXCT only scans 30 μ m in the vertical direction of the capillaries (meanwhile Syn- μ CT and Lab- μ CT scan 1000 μ m). With large alite grains of about 20 μ m, scanning 30 μ m in the vertical direction is limited (although 160 μ m was imaged in the transversal section of the capillary which allowed to scan many particles. Indeed, to image a larger volume, vertical dimension of about 60-100 μ m would require about 7-9 hours per scan which may not allow to accurately study the hydration as important changes can occur during the acquisition time.

We have clarified this by adding the following set of sentences to the revised version on page 12: "... Finally, it should be noted that the scanned length in the vertical direction, 30 μ m, for PXCT is limited taking into account that some alite grains with sizes of 20 μ m, or slightly larger, are present in PC cements. This was mitigated by imaging 160 μ m in the transversal direction. This type of experiment will benefit from imaging cylindrical volumes with 60-100 μ m of height. However, with the current experimental procedure, this would lead to acquisition times larger than 7-9 hours and therefore changes due to hydration could take place during an acquisition. Procedures for faster data collection are being explored and some are discussed in the next section."

2.20. What is the UCP volume mentioned in page 10?

2.20. Author reply: This is the total volume of the anhydrous cement particles at 19 h as determined from PXCT. We now realize the sentence was not fully clear and we have replaced on page 12 "... The groups contained 1117, 204, 61 and 20 particles, respectively, and the corresponding percentages with respect to the UCP volume were 5.4, 12.9, 21.5 and 60.2%." by "... The groups contained 1117, 204, 61 and 20 particles, respectively. The corresponding volume percentages with respect to the overall anhydrous cement particle volume at this hydration age, were 5.4, 12.9, 21.5 and 60.2%."

2.21. How is it possible to link the results obtained from 2D figures 5 and S13 to S18 to the ones obtained with the 3D segmentation? Are the results giving the same spatial dissolution rate?

2.21. Author reply: We thank the reviewer for allowing us to elaborate on this important result from the segmentation quantitative analysis point of view. Based on quantitative analysis derived from ML segmentation of the PXCT datasets (for the C_3S/C_2S class, dark brown colour code in the 3D visualization of Fig. 7), it is possible to derive an average spatial dissolution rate. We used mathematical morphology tools to retrieve the outer layer of the 19 h hydration and 47 h hydration segmented grain. Then, we computed the average distance between these outer layers for each grain, giving a mean value of 1.92 pixels (i.e. ~ 13 nm/h). This value is smaller than that obtained from 2D analysis for alite, 25 nm/h. However, it should be noted that in the segmentation analysis, alite and belite were classified together and therefore, the obtained spatial dissolution rate is underestimated as belite does not dissolve at early ages. Table S5, from the Rietveld analysis, indicates that the amount of belite is approximately half of that of alite during this stage. Therefore, the spatial dissolution rate can be corrected. The value, only for alite, would be close to 13/0.67 or 19 nm/h. This rate agrees relatively well with that obtained from the 2D analysis of 22 measurements, 25 nm/h, given the number of approximations which took place for both calculations. Now in the revised version on page 7: "... Moreover, this spatial dissolution rate can also be estimated from the segmentation results presented in the next subsection. Based on the quantitative analysis derived from Machine Learning (ML) segmentation of the PXCT datasets (for the C_3S/C_2S class, dark brown colour code in the 3D visualization of Fig. 7), it is possible to derive an average spatial dissolution rate. Mathematical morphology tools were used to retrieve the outer layer of the segmented grains at 19 and 47 h. Subsequently, the average distance between these outer layers was computed for each grain, giving a mean value of 1.92 pixels (i.e. ~ 13 nm/h). This value is smaller than that obtained from the analysis performed in 2D slices for alite, 25 nm/h. However, it should be noted that in the segmentation calculation, alite and belite were classified together and therefore, the obtained spatial dissolution rate is underestimated as belite does not dissolve at early ages. Table S5, Rietveld analysis results, indicates that the amount of belite is half of that of alite during this stage. Therefore, the spatial dissolution rate can be corrected. The alite spatial dissolution rate should be close to 13/0.67 or 19 nm/h. This value agrees relatively well with 25 nm/h, obtained from 22 measurements in 2D slices, given the number of approximations which took place for both calculations...."

2.22. Can you specify what you mean by quantification in this sentence: "Etch-pit growth rate, ~ 40 nm/h, and coalescence have also been measured but better spatial resolution is required for etch-pit quantification"?

2.22. Author reply: This point is related to that about etch-pit evaluation, 2.15. We acknowledge that the wording was somehow misleading. The etch-pit growth rate was/is an estimation, not an accurate measurement. In fact, due to this, we decided not to report a number in the abstract. Thus, "Etch-pit growth rate, ~ 40 nm/h, and coalescence have also been measured but better spatial resolution is required for etch-pit quantification" has been replaced by : "... The alite etch-pit growth rate between 19 and 47 h has been estimated as ~ 40 nm/h, which

decreases to ~7 nm/h in the 47 to 93 h interval. Moreover, etch-pit coalescence, the merging of different branches, has also been visually observed. However, better spatial resolution is required for a thorough etch-pit growth rate quantification.”

2.23. At the end of the “ implications and outlook” paragraph it is written that “ The current spatial resolution of in situ near-field PXCT, ~370 nm, can be improved by increasing the number of projections, without larger acquisition times” which, from my knowledge, is false. While in the supplementary info section it is reported and especially pointed out that “The resolution obtained, see subsection dedicated to the spatial resolution, was limited by the number of projections, which was chosen to have reasonable scan times”.

2.23. Author reply: Clearly, as previously stated there was a contradiction. This is very much related to point 1.1 above. The way to have better resolution (acquiring more projections) without increasing the dose too much (to avoid radiation damage) is to decrease the acquisition times. As stated in the answer to point 1.1-c the approach would be to collect data with faster acquisition times (worse signal-to-noise ratio) and to use ML for reconstruction and denoising of datasets. Alternatively, sparsity techniques can be coupled to PXCT, which is being implemented at cSAXS beamline. The text was revised as stated in point 1.1, above.

2.24. Is the sample perfectly stable from one measurement to another? Is there a need to “realign” the volume from one-step to another? Is there a need for volume registration? If yes, how has it be done?

2.24. Author reply: This is an important question that it requires a detailed answer. We acknowledge that the original submission did not contain the requested details. Furthermore, as there are three imaging modalities (ptychography, synchrotron microtomography and laboratory microtomography) the answer depends upon the employed technique. This has been addressed by adding the following information in the S.I. material:

“... Initially, the re-alignment of the data, when needed must be detailed.

For the PXCT, the capillary position was very accurate, as the capillary/holder system was mounted from the tray storage to the sample stage by the fLOMNY gripper (robot). Hence, the angular orientation of the sample was maintained. The field of view of the sample was aligned carefully based on features visible in the 2D projections. The scanned regions with time were consistent within a few voxels and therefore no alignment between different acquisitions was required.

For the Syn- μ CT, a mark was drawn on the sample holder and sample stage for the incident beam to minimise the initial incidence angular position variability. Before each scan, a projection was acquired as a reference for the next one in order to scan the same region. A minor manual registration was required, mostly rotations around x- and y-axes.

For the Lab- μ CT, manual registration was required to align the different acquisitions. The process is described next. The capillary was considered as a cylinder and we manually made the cylinders vertical and centred in the reconstructed volume. The remaining rotation around the z-axis was visually done by superimposing distinguishable landmarks in the corresponding images.

Reviewer #3 (Remarks to the Author):

Major revision is needed.

This paper presents a method to observe the hydration process of cement using 4D nanoimaging. The idea is novel, but there are some problems:

Author reply: Many thanks.

3.1. The literature should be updated, more literature should be in recent three years.

3.1. Author reply: We thank the reviewer for this comment. We have included five new references (2020-2023), just to respect the maximum number of references, which in the guideline for submission is stated as 70. The revised version of this manuscript had already 69 references. Now in the introduction of the revised version “... In the last three years, important advances have been reported including: i) the automated correction for the movement of suspended particles at very early ages³⁴ which allowed to follow *in situ* PC hydration after water mixing³⁵; ii) to follow the fast dissolution of plaster and the precipitation of gypsum³⁶; iii) the simultaneous use of neutron and laboratory X-ray tomographies for *in situ* studying the microstructural changes of PC mortars on moderate heating³⁷; and iv) the measurement of alite particle dissolution using fast synchrotron nano X-ray computed tomography^{38,39}” The new references added are the followings (reference 35 was already included in the original submission):

(34) Vigor, J. E., Bernal, S. A., Xiao, X. & Provis, J. L. Automated correction for the movement of suspended particulate in microtomographic data. *Chem. Eng. Sci.* **223**, 115736 (2020).

(36) Seiller, J. *et al.* 4D in situ monitoring of the setting of a plaster using synchrotron X-ray tomography with

high spatial and temporal resolution. *Constr. Build. Mater.* **304**, 124632 (2021).

(37) Cheikh Sleiman, H., Tengattini, A., Briffaut, M., Huet, B. & Dal Pont, S. Simultaneous x-ray and neutron 4D tomographic study of drying-driven hydro-mechanical behavior of cement-based materials at moderate temperatures. *Cem. Concr. Res.* **147**, 106503 (2021).

(38) Li, X. *et al.* Direct observation of C3S particle dissolution using fast nano X-ray computed tomography. *Cem. Concr. Res.* **166**, 107097 (2023).

(39) Li, X., Hu, Q., Robertson, B., Tyler Ley, M. & Xiao, X. Direct observation of C3S particles greater than 10 μm during early hydration. *Constr. Build. Mater.* **369**, 130548 (2023).

3.2. In the Introduction, the disadvantages of the references should be summarized clearly to emphasize the importance of this work. The difficulties of 4D nanoimaging also need to be summarized.

3.2. Author reply-a: Concerning the disadvantages of the published work, we thank the reviewer since this helps to focus on the work we present in this manuscript. This was already treated in the introduction but we acknowledge that not with the required depth. Now in the revised version: "... In particular, hard X-ray synchrotron microtomography has not the required submicrometer spatial resolution neither sufficient component contrast^{35,40}, hard X-ray synchrotron nanotomography has not the required contrast between the components to be able to classify the hydrates^{38,39} and soft X-ray synchrotron nanotomography has the contrast but it requires very large w/c ratios and very small fields of view which does not allow the hydrates to growth in relevant condition (i.e. confined space with low water-cement ratios)³²."

3.2. Author reply-b: We think that the difficulties were already detailed in the introduction by stating the four (stringent) requirements that should be met simultaneously. To refer this to 4D imaging in a more obvious way, we have rephrased now this part on page 2: "... However, none of these 4D imaging works combine the stringent four requirements needed for carrying out relevant contributions to the understanding of the mechanism(s) of Portland cement hydration at early ages: (i) water to cement mass ratio (w/c) close to 0.50, (ii) submicrometer spatial resolution, (iii) good contrast to be able to identify the different evolving components (more than eight), and (iv) relatively large scanned volume to allow hydration to progress with appropriate particle sampling, the particle sizes of commercial PCs have $D_{v,50} \in 10\text{-}20 \mu\text{m}$." We prefer not to reiterate this message.

3.3. In Fig.7, the time selection is very strange."19h","47h","93h". what is the reason. The title of this paper focuses on the early hydration. why not select some early hydration points? for example, Non-contact multiple-frequencies AC impedance instrument for cement hydration based on a high-frequency weak current sensor.

3.3. Author reply: The selection of the hydration time was given by the availability of synchrotron beamtime that was shared with other nanotomographic study. On the other hand, there are several techniques that can help to characterise the overall water porosity evolution with time such as "Non-Contact Multiple-Frequency AC Impedance Instrument for Cement Hydration Based on a High-Frequency Weak Current Sensor; doi: 10.3390/act12010026" but they lack the submicrometer spatial resolution that is the focus of this work.

3.4. This paper is interesting, cement pastes are multi-phase materials. Some phases are overlapped with others in some localized regions. Some nano-scale C-S-H and CH may be mixed together. It is said that the intrinsic size of C-S-H is about 5 nm (doi.org/10.3390/fractalfract5020047). Therefore, in Fig.4 nanoimaging, C-S-H may not be identified in the scale bar 5 μm . Therefore, at least, the typical size of each phase during the cement hydration should be listed out from literature, so that reader can compare comprehensively.

3.4. Author reply: We fully agree with the reviewer's statement. This is also linked to the partial volume effects highlighted by reviewer 2 in his/her remark 2.13. We have clarified this by adding the following set of sentences to the text: "...It should be noted that individual C-S-H nanoparticles have sizes close to 5 nm^{12,55} much smaller than the spatial resolution of this work, i.e. $\sim 250 \text{ nm}$. Therefore, the C-S-H regions analysed here very likely contain other components like gel and capillary water porosities and interspersed calcium hydroxide. On the other hand, ettringite and portlandite particles have larger sizes, usually ranging 1-5 μm , and they can be imaged in the present work.³ In any case, partial volume effect (the presence of components contributing below the spatial resolution of the measurements) is always taking place in cement pastes as some hydrates may have quite small particle sizes.¹⁷" We have also added the suggested reference:

(55) Tang, S. *et al.* Structure, fractality, mechanics and durability of calcium silicate hydrates. *Fractal Fract.* **5**, 47 (2021).

3.5. As we know, some commercial PCs have some mineral admixtures. What about the XRF results of these two commercial PCs in this work.

3.5. Author reply: Indeed, most commercial PCs have limestone in addition to the gypsum used as set regulator, this is also linked to the 3.7 point, below. The employed PCs are type I where only gypsum and limestone are added to the clinker and their contents lower than 5 wt%. The XRF, and the LoI, data were given in Table S1.

3.6. Fig. 3. Left figure is terrible. We can not identify which is which. It is only black and while.

3.6. Author reply: We have made an effort to clarify the message but the bottom-line feature cannot change: the relatively poor spatial resolution and contrast of synchrotron (propagation-based phase-contrast) microtomography when compared to ptychographic nanotomography. When inspecting the fine details at the same (high) magnification, synchrotron microtomography shows the relatively coarse voxel size with limited contrast. Figure 3 caption has been revised to detail the new views. Now in the revised version:

Fig. 3. Comparison of phase-contrast synchrotron tomography and ptychographic X-ray computed tomography. **a**, Selected orthoslices at 19 h for (top) Syn- μ CT [voxel size: 650 nm, total scanned volume: $5.25 \cdot 10^8 \mu\text{m}^3$, overall acquisition time: 5 min], and (bottom) PXCT [voxel size: 186.64 nm, total scanned volume: $8.15 \cdot 10^5 \mu\text{m}^3$, overall acquisition time: 3 h, 55 min]. **b**, enlarged views of the highlighted regions (purple squares) in **a**, in order to illustrate the level of detail that can be observed with these imaging modalities. Every voxel in Syn- μ CT image starts to be evident. **c**, Further enlarged views to illustrate the maximum level of detail that can be observed. (Top) The Syn- μ CT image shows whitish particles (anhydrous cement particles) surrounded by hydrates (greyish voxels) which are highlighted by red arrows. (Bottom) The PXCT data, at the same magnification, shows a much higher level of detail. The C-S-H gel shells surrounding the alite particles are clearly visible (pink arrows). There is a water gap between the shell and the alite grain due to the inward dissolution of alite. Moreover, etch-pits on the surfaces of the alite particles are also visible (blue circles). The highest spatial resolution and better contrast of PXCT data allow visualizing submicrometre features of the dissolution-precipitation processes which are not visible in propagation-based Syn- μ CT. Conversely, propagation-based Syn- μ CT permits to scan much larger volumes in much smaller acquisition times, showing the complementary nature of both techniques.

3.7. From Fig.7, the samples used in PXCT have serious carbonation. However, other samples do not have such carbonation. Besides, in Fig.7, how do authors identify the different kinds of C-S-H gels. Is it from AFM test according to porosity or NMR according to silicate calcium chain (doi.org/10.1016/j.conbuildmat.2020.118807 & doi.org/10.1016/j.measurement.2021.110019)

3.7. Author reply-a: Related to carbonation. We do not agree with this statement. The samples have not serious carbonation. It is noted that the employed cements are commercial and these materials have normally 4-5 wt% of limestone addition. This was already shown in the cement mineralogical analyses, see Table S2, where PC-52.5 and PC-42.5 have 2.8 and 3.7 wt% of crystalline calcite, respectively. This is now clarified in the caption of Figure 7 by adding: "... This calcite comes very likely from the limestone addition to the Portland cement as quantified in the anhydrous material, see Table S2."

3.7. Author reply-b: We are aware that there are different C-S-H from the nano-mechanical point of view with different gel pore water contents. Unfortunately, with the spatial resolution and contrast currently available, we cannot distinguish them. From synchrotron microtomography we can distinguish hydrates with high mass/electron densities (portlandite, calcite, high-density C-S-H) from low density compounds (ettringite, AFm-type phases and low-density C-S-H). PXCT allows to distinguish calcite but it does not allow to distinguish high

density and low density C-S-H. This is now clarified in the caption of Figure 7 by adding: "... It is noted that with the quality of the data reported in this study (spatial resolution and electron density contrast), it is not possible to disentangle low-density from high-density C-S-H".

3.8. In Fig.6, authors claim they can monitor the evolution of water porosity (dark-grey) to air porosity (black). Does it mean that water can be distinguished by near field PXCT? As I know, only neutron can examine the presence of water.

3.8. Yes, PXCT can distinguish water from air, if the measurements are done to yield quantitative electron densities. This requires thin specimens surrounded by air. This is known since 2012: "Quantitative x-ray phase nanotomography, doi: 10.1103/PhysRevB.85.020104", where a sensitivity of about $0.02 \text{ e}^{-}\text{\AA}^{-3}$ was demonstrated for the electron density contrast. The electron density of water is $0.33 \text{ e}^{-}\text{\AA}^{-3}$, larger than that of air, i.e. $0.00 \text{ e}^{-}\text{\AA}^{-3}$. This is clarified in the revised version "... It is underlined that PXCT readily distinguishes air and water porosities because of their difference in electron densities, 0 and $0.33 \text{ e}^{-}\text{\AA}^{-3}$, respectively; when the phase retrieval is carried out quantitatively⁵⁴".

Obviously, we fully agree that neutron imaging is another technique that can disentangle water from air porosities. The key advantage of neutron imaging is clear, it can scan large volumes. However, it must also be noted that at a much worse spatial resolution compared to PXCT.

We have clarified this in the S.I. by adding "... PXCT provides 3D maps of the electron density of the specimen with quantitative contrast, the sensitivity being about $0.02 \text{ e}^{-}\text{\AA}^{-3}$.¹⁶ For attaining quantitative electron densities, the entire specimen must be included in the field of view, containing some empty space around it, which was the case in our measurements. Therefore, it is possible to easily distinguish air and water regions in the specimen, which have electron densities of 0.00 and $0.33 \text{ e}^{-}\text{\AA}^{-3}$, respectively. Obviously, neutron imaging is the standard technique to disentangle water from air porosities. A key advantage of neutron imaging is its ability to scan large volumes. However, it must also be noted that at a much worse spatial resolution compared to PXCT."

3.9. In Fig.8, PXCT test results seem that C_4AF content does not change much. It is not reasonable. Common sense tells us that the hydration rate is $C_3A > C_3S > C_4AF > C_2S$.

3.9. Author reply: The reviewer is right to point out a) the well-known hydration rate sequence, $C_3A \sim C_3S > C_4AF > C_2S$; and b) that C_4AF does not change much in our study. However, we do not consider this observation unreasonably for two reasons: i) C_4AF reactivity starts to take place mainly after about 2 days of hydration and our last measurement took place at 4 days (i.e. 93 h); and ii) the w/c ratio in the scanned volume was about 0.40. A low w/c ratio means less water available for the hydration of the phases with the slower kinetics, i.e. C_4AF . We have clarified this by adding the following statement in the figure caption. "... PXCT data shows that C_4AF hydrates little up to 93 h. This is likely due to the low w/c ratio in the scanned volume and its slow hydration rate."

Reviewer #4 (Remarks to the Author):

A superbly presented, detailed study of cement hydration using various X-ray methods. I would recommend a review by someone expert in the cement hydration field, since I cannot comment on the impact of the results from the point of view of the application. What I can say is that the ptychographic imaging is a significant real-world demonstration of the near-field approach and, to my mind, represents the current state-of-the-art. I have no issue recommending the article for publication, subject only to minor improvement of the English - which although always comprehensible could do with a proof-read.

Author reply: Many thanks

REVIEWERS' COMMENTS

Reviewer #1 (Remarks to the Author):

The author has replied to all the issues and revised the manuscript accordingly. I recommend it for publication.

Reviewer #2 (Remarks to the Author):

First of all, I would like to thank the authors for their very detailed and precise answers. I acknowledge the work they have done to improve the manuscript and precise the methods used for the quantitative analysis of the images. I would recommend adding graphs detailing the large variability in the results of the etch-pit growth rate in the SI. I am happy to reconsider my previous decision and support this work for publication in Nature communications.

Reviewer #3 (Remarks to the Author):

Major revision is needed.

- 1) In Table S5, the content of Cc increases from 2.0 to 3.6 at 96h. obviously, the serious carbonation occurs, see the review comments 3.7
- 2) In Table S7, data of AFm and C-A-S-H are missing. As Ca/Si ratio of C-S-H is very close to 1.7, not 1.8. Do authors have data related to C-S-H with Ca/Si 1.7. C-S-H gels are glassy phases so their values are within a certain range in Table S7. What is the range? The electron density of C-S-H and CH are close to each other. In most cases, the hydrating pastes, C-S-H and CH are mixed together with porosity, and hard to be distinguished. Once C-S-H and CH can not be distinguished, 4D nanoimaging of hydration can not be achieved because they are occupied up to 80% volume of hydrates.
- 3) Some key points should be clarified. Raw clinker of cement is mixture. The microstructure of pastes includes time-evolving multi-phases with different multi-scale(nano-, meso-, and micro). Therefore, if PXCT enables to distinguish the amount and distribution of different phases, these phases should have "distinct" features in electron density derived from PXCT. Obviously, single test or measurement can not achieve this objective. Therefore, other test results and literature results should be carried out and compared to support the evidences in the manuscript. Please the review comment 3.7.

Response to reviewers document. Reference NCOMMS-22-47460A (second revision)

Title. “4D nanoimaging of early age cement hydration”

Reviewer #1 (Remarks to the Author):

The author has replied to all the issues and revised the manuscript accordingly. I recommend it for publication.

Author reply: Many thanks.

Reviewer #2 (Remarks to the Author):

First of all, I would like to thank the authors for their very detailed and precise answers. I acknowledge the work they have done to improve the manuscript and precise the methods used for the quantitative analysis of the images. I would recommend adding graphs detailing the large variability in the results of the etch-pit growth rate in the SI. I am happy to reconsider my previous decision and support this work for publication in Nature communications.

Author reply-1: Many thanks.

Author reply-2: Concerning the etch-pit growth rate, we can only agree. Therefore, we have prepared a new figure, Fig. S13 with two particles showing the evolution of ten etch-pits. The remaining figures have been renumbered. Figure S13 and its caption are reproduced below.

Figure S13. Overlay of the (2D-projected) alite segmented pixels during the hydration process to show the large variability in the growth rates of the etch-pits. **a**, five etch-pits corresponding to Figure 4 of the manuscript. **b**, five etch-pits corresponding to the Figure S11 of this S.I. Pale-blue arrows show size changes from 19h to 47h, meanwhile white arrows display the changes from 47h to 93h.

Reviewer #3 (Remarks to the Author):

Major revision is needed.

3.1. In Table S5, the content of Cc increases from 2.0 to 3.6 at 96h. obviously, the serious carbonation occurs, see the review comments 3.7.

Author reply: The initial PC had an average calcite content of 2.8 wt%, see Table S2, that when diluted with the water resulted in a calcite content of 2.0 wt% (value at t=0). For the large capillary used in the laboratory micro-CT & LXRPD study, $\phi=1.0$ mm, the calcite content increased to about 3.6 wt% at 96 h, i.e. the calcite amount increased 1.6 wt%. This is now explicitly acknowledged in the footnote of Table S5. However, the paste within the much thinner capillary, $\phi=0.2$ mm, used in the PXCT study did not show a significant carbonation. Now in the revised version of S.I., a footnote of Table S5:

“# The calcite content increased from 2.0 wt% at t=0 to 3.6 wt% at 96 h, highlighting a significant carbonation of the paste within this large capillary, i.e. 1 mm of diameter. The thinner capillary used in the PXCT study, i.e. 0.2 mm of diameter, did not show a measurable conversion of CH to Cc, see below. Carbonation of a cement paste has been previously measured by PXCT, when its extension was significant.²²”

We have added new reference (22) in the S.I.: Trtik, P., Diaz, A., Guizar-Sicairos, M., Menzel, A. & Bunk, O. Density mapping of hardened cement paste using ptychographic X-ray computed tomography. *Cem. Concr. Compos.* **36**, 71–77 (2013)

3.2. In Table S7, data of AFm and C-A-S-H are missing. As Ca/Si ratio of C-S-H is very close to 1.7, not 1.8. Do authors have data related to C-S-H with Ca/Si 1.7. C-S-H gels are glassy phases so their values are within a certain range in Table S7. What is the range? The electron density of C-S-H and CH are close to each other. In most cases, the hydrating pastes, C-S-H and CH are mixed together with porosity, and hard to be distinguished. Once C-S-H and CH can not be distinguished, 4D nanoimaging of hydration can not be achieved because they are occupied up to 80% volume of hydrates.

Author reply-1: We agree that data for AFm and C-A-S-H were missed in Table S7. Thus, the expected data for AFm and hemiacarbonate (Hc) phases are now included. We cannot include data for C-A-S-H as this component, with aluminium content larger than ~5 %, mainly forms in the pozzolanic reaction which is not the subject of this investigation.

Author reply-2: Concerning the variable Ca/Si ratio. C-S-H gel from PC hydration, without supplementary materials addition, shows variable Ca/Si ratios close to 1.7-1.8. This is now explicitly stated in the revised version of the manuscript “It should also be noted that the employed stoichiometry for C-S-H gel, i.e. $(\text{CaO})_{1.80}(\text{SiO}_2)(\text{H}_2\text{O})_{4.0}$,^{56,57} is an assumption and slightly smaller Ca/Si ratios, ~1.70, have also been reported.^{12,58}”. Three new references (56-58) are added: (56) Cuesta, A. *et al.* Local structure and Ca/Si ratio in C-S-H gels from the hydration of blends of tricalcium silicate and silica fume. *Cem. Concr. Res.* **143**, 106405 (2021); (57) Zhu, X. & Richardson, I. G. Morphology-structural change of C-A-S-H gel in blended cements. *Cem. Concr. Res.* **168**, 107156 (2023); (58) Duque-Redondo, E., Bonnaud, P. A. & Manzano, H. A comprehensive review of C-S-H empirical and computational models, their applications, and practical aspects. *Cem. Concr. Res.* **156**, 106784 (2022).

We have calculate the values for $(\text{CaO})_{1.7}(\text{SiO}_2)(\text{H}_2\text{O})_4$ and compared them with those of $(\text{CaO})_{1.8}(\text{SiO}_2)(\text{H}_2\text{O})_4$. For the first average stoichiometry, the electron density and μ values would be $0.66 \text{ e}^{-\text{\AA}^{-3}}$ and 102 cm^{-1} . The corresponding values for the second component are $0.66 \text{ e}^{-\text{\AA}^{-3}}$ and 104 cm^{-1} . We do not report the values for $(\text{CaO})_{1.7}(\text{SiO}_2)(\text{H}_2\text{O})_4$ in Table S7 as they are, as expected, very close to those of $(\text{CaO})_{1.8}(\text{SiO}_2)(\text{H}_2\text{O})_4$.

Author reply-3: Concerning C-S-H and CH electron density values. Indeed, the electron density values of these components are close and in some regions, they are interspersed. We could disentangle these components by nano-imaging in most regions. However, the presence of partial volume effect was explicitly acknowledged in the manuscript and reproduced here: “... In any case, partial volume effect (the presence of components contributing below the spatial resolution of the measurements) is always taking place in cement pastes as some hydrates may have quite small particle sizes.¹⁷”

3.3. Some key points should be clarified. Raw clinker of cement is mixture. The microstructure of pastes includes time-evolving multi-phases with different multi-scale(nano-,meso-, and micro). Therefore, if PXCT enables to distinguish the amount and distribution of different phases, these phases should have “distinct” features in electron density derived from PXCT. Obviously, single test or measurement can not achieve this objective. Therefore, other test results and literature results should be carried out and compared to support the evidences in the manuscript. Please the review comment 3.7.

Author reply-1: The reviewer is right in pointing out that the cements are mixtures. This was detailed in the S.I. by reporting their mineralogical analyses. However, this is now clarified in the main text by adding: "... As reported in Table S2, the anhydrous cements are mixtures with more than eight crystalline phases."

Author reply-2: PXCT allows distinguishing components within the hydrating cement paste but it has limitations that are detailed in the manuscript. The two main limitations are: (i) the current spatial resolution, i.e. 250 nm. Clearly, component interspersed at smaller length scales cannot be disentangled, leading to partial volume effects. (ii) The electron density contrast is about 0.02-0.04 e⁻Å⁻³ and components with electron densities closer than ~0.04 e⁻Å⁻³ cannot be separated. However, to use additional techniques to *in situ* study the hydration of these cements (in addition to the already employed ones: calorimetry, laboratory powder diffraction and laboratory microtomography) will be the subject of further works, but this is not within the scope of the present investigation.